# Unifying and Enhancing Graph Transformers via a Hierarchical Mask Framework

**Yujie Xing**[1], **Xiao Wang**[2*], **Bin Wu**[1], **Hai Huang**[1], **Chuan Shi**[1*]
[1]Beijing University of Posts and Telecommunications, China
[2]Beihang University, China
{yujie-xing,wb789,hhuang,shichuan}@bupt.edu.cn, xiao_wang@buaa.edu.cn

## Abstract

Graph Transformers (GTs) have emerged as a powerful paradigm for graph representation learning due to their ability to model diverse node interactions. However, existing GTs often rely on intricate architectural designs tailored to specific interactions, limiting their flexibly. To address this, we propose a unified hierarchical mask framework that reveals an underlying equivalence between model architecture and attention mask construction. This framework enables a consistent modeling paradigm by capturing diverse interactions through carefully designed attention masks. Theoretical analysis under this framework demonstrates that the probability of correct classification positively correlates with the receptive field size and label consistency, leading to a fundamental design principle: *An effective attention mask should ensure both a sufficiently large receptive field and a high level of label consistency*. While no single existing mask satisfies this principle across all scenarios, our analysis reveals that hierarchical masks offer complementary strengths—motivating their effective integration. Then, we introduce M$^3$Dphormer, a **M**ixture-of-Experts based Gra**ph** Transf**ormer** with **M**ulti-Level **M**asking and **D**ual Attention Computation. M$^3$Dphormer incorporates three theoretically grounded hierarchical masks and employs a bi-level expert routing mechanism to adaptively integrate multi-level interaction information. To ensure scalability, we further introduce a dual attention computation scheme that dynamically switches between dense and sparse modes based on local mask sparsity. Extensive experiments across multiple benchmarks demonstrate that M$^3$Dphormer achieves state-of-the-art performance, validating the effectiveness of our unified framework and model design. The source code is available for reproducibility at: https://github.com/null-xyj/M3Dphormer.

## 1 Introduction

As a fundamental data structure, graphs have been widely used to model complex and diverse interactions in real-world systems, such as social networks and brain networks. To learn high-quality node representations, a variety of Graph Neural Networks (GNNs) have been proposed [23, 38, 16]. However, their performance is inherently constrained by the message-passing mechanism, which imposes a strong locality inductive bias. Inspired by the success of Transformers [37] across various machine learning domains [12, 13], adapting Transformer architectures to graphs has emerged as a promising direction, owing to their strong capability to model interactions over a broader range.

Graph Transformers (GTs) leverage the core component of the Transformer architecture, Multi-Head Attention (MHA), to adaptively model diverse node interactions and learn expressive representations. One prominent line of research treats the entire graph as fully connected and applies attention mechanisms to capture pairwise node dependencies [42, 43, 11]. Another line constructs a token

---

*Corresponding authors

sequence for each node, typically via node sampling or feature aggregation, and adopts a Transformer to capture multi-scale interactions. [5, 15, 41]. Besides, several studies leverage graph partitioning to enable efficient interaction modeling and learn high-quality representations. [18, 20, 44].

While many GTs have been proposed, they often rely on intricate architectural design tailored to specific types of node interactions, which limits their ability to model other important interactions. This raises a natural question: *Does there exist a unified perspective of GTs that allows for flexible modeling of diverse node interactions?* To address this question, we propose a unified hierarchical mask framework, developed through a thorough analysis of existing GTs. Our analysis reveals that various GT architectures inherently model interactions at different levels—local, cluster, and global—corresponding to the hierarchical organization of relational patterns in graphs. Furthermore, we find that these hierarchical interactions can be uniformly modeled through the design of appropriate attention masks, and that many existing GTs can be interpreted as implicitly corresponding to specific masks. This unified perspective reveals an underlying equivalence between model architecture and mask construction, offering a more flexible approach to GT design.

Under this unified framework, diverse node interactions can be modeled in a consistent manner through the construction of appropriate attention masks, avoiding the need to design intricate network architectures as in traditional methods. Moreover, it facilitates theoretical analysis, proving that both the lower and upper bounds of the probability of correct classification correlate positively with the size of the receptive field and the degree of label consistency. This leads to a fundamental design principle for attention masks: *An effective attention mask should ensure a sufficiently large receptive field and a high level of label consistency.* Further analysis of masks derived from existing GTs shows that no single mask consistently satisfies this principle across all scenarios. However, hierarchical masks exhibit complementary strengths in node classification, suggesting that integrating multi-level masks provides a natural and effective way to adhere to this principle.

Then, we conduct experiments on real-world datasets to investigate the effectiveness of combining masks across multiple levels. Specifically, we construct three GTs, each designed to capture a single level of interaction—local, cluster, or global—using a corresponding attention mask. We then apply three ensemble strategies to integrate their outputs: Mean, Max, and an idealized Oracle. The node classification results show that: 1) The Oracle strategy significantly outperforms all other models, demonstrating the potential of comprehensively leveraging multi-level interactions. 2) Naive ensemble methods (Mean and Max) often underperform than the best individual-mask model, highlighting a core challenge in effectively integrating hierarchical information. In addition, the excessive memory usage on medium-scale graphs further reveals a key efficiency challenge in GTs.

In this paper, we propose M$^3$Dphormer, a novel **M**ixture-of-Experts based Gra**ph** Transf**ormer** with **M**ulti-Level **M**asking and **D**ual Attention Computation. Specifically, M$^3$Dphormer employs three theoretically grounded attention masks for comprehensive modeling of hierarchical interactions, including local, cluster, and global associations. To effectively integrate information across these interaction levels, we design a bi-level expert routing mechanism, where each expert is a multi-head attention module associated with a specific mask. Furthermore, a dual attention computation strategy is introduced to enhance scalability and computational efficiency.

Our main contributions are summarized as follows:

- We propose a unified hierarchical mask framework that reveals an underlying equivalence between model architecture and attention mask construction, enabling diverse node interactions to be consistently modeled through carefully designed masks.

- Theoretical analysis within this framework reveals that the probability of correct classification is positively correlated with both the receptive field size and label consistency. This leads to a guiding principle for designing attention masks: an effective mask should ensure a sufficiently large receptive field and a high level of label consistency.

- We propose M$^3$Dphormer, a novel Graph Transformer that captures hierarchical interactions comprehensively and efficiently through multi-level masking, bi-level expert routing, and dual attention computation, thereby adhering to the proposed design principle.

- We perform extensive experiments on 9 benchmark datasets, showing that M$^3$Dphormer consistently outperforms 15 strong baselines, demonstrating its effectiveness.

## 2 Preliminary

We denote an attributed graph as $\mathcal{G} = (\mathcal{V}, \mathcal{E}, \mathbf{X})$, where $\mathcal{V} = \{0, 1, \cdots, N-1\}$ is the set of $N$ nodes, $\mathcal{E}$ is the set of $E$ edges, and $\mathbf{X} = [\mathbf{x}_u] \in \mathbb{R}^{N \times d_{in}}$ is the node feature matrix, with $\mathbf{x}_u \in \mathbb{R}^{d_{in}}$ representing the $d_{in}$-dimensional feature vector of node $u$. The adjacency matrix is denoted as $\mathbf{A} = [a_{uv}] \in \{0, 1\}^{N \times N}$, where $a_{uv} = 1$ if there exists an edge from node $u$ to node $v$, and $a_{uv} = 0$ otherwise. For node classification, we define the set of labels as $\mathcal{Y}$, and represent the node labels by $\mathbf{Y} = [\mathbf{y}_u] \in \{0, 1\}^{N \times |\mathcal{Y}|}$, where $\mathbf{y}_u \in \{0, 1\}^{|\mathcal{Y}|}$ is the one hot label of node $u$. The whole node set can be divided into the training set $\mathcal{V}_{train}$, the valid set $\mathcal{V}_{valid}$, and the test set $\mathcal{V}_{test}$.

**Graph transformers:** The core component of GTs is the MHA, formulated as follows:

$$\text{head}_i(\mathbf{H}, \mathbf{M}) = \text{Softmax}\left(\text{Mask}\left(\hat{\mathbf{A}}^{(i)}, \mathbf{M}\right)\right)\mathbf{V}^{(i)}, \quad i = 1, \ldots, H$$

$$\hat{\mathbf{A}}^{(i)} = \frac{\mathbf{Q}^{(i)}\mathbf{K}^{(i)\top}}{\sqrt{d_h}}, \quad \mathbf{Q}^{(i)} = \mathbf{H}\mathbf{W}_Q^{(i)}, \quad \mathbf{K}^{(i)} = \mathbf{H}\mathbf{W}_K^{(i)}, \quad \mathbf{V}^{(i)} = \mathbf{H}\mathbf{W}_V^{(i)} \tag{1}$$

$$\text{Mask}(\hat{\mathbf{A}}^{(i)}, \mathbf{M}) = \begin{cases} \hat{\mathbf{A}}_{u,v}^{(i)}, & \text{if } \mathbf{M}_{u,v} = 1 \\ -\infty, & \text{if } \mathbf{M}_{u,v} = 0 \end{cases}$$

Here, $\mathbf{H} \in \mathbb{R}^{N \times d}$ denotes the input representations, where $N$ is the number of nodes and $d$ is the representation dimension. $\mathbf{W}_Q^{(i)}, \mathbf{W}_K^{(i)}, \mathbf{W}_V^{(i)} \in \mathbb{R}^{d \times d_h}$ are trainable projection matrices specific to the $i$-th head, and $d_h$ is the hidden dimension of each head. The attention score matrix $\hat{\mathbf{A}}^{(i)}$ is masked by $\mathbf{M} \in \{0, 1\}^{N \times N}$, where $\mathbf{M}_{u,v} = 1$ indicates the validity of the attention from node $u$ to node $v$. During masking, attention scores corresponding to invalid positions are set to $-\infty$, ensuring that their contribution becomes zero after the Softmax operation. The outputs of all heads are concatenated to produce the final output: $\text{MHA}(\mathbf{H}, \mathbf{M}) = \text{Concat}(\text{head}_1, \ldots, \text{head}_H)$.

## 3 Revisiting GTs through a unified hierarchical mask framework

Interactions in graphs typically exhibit a hierarchical organization, including local connectivity, cluster relations, and global associations. Each level provides essential information for effective graph representation learning. An illustrative example for the importance of hierarchical interactions is provided in Figure 1, where node labels are indicated by different colors. Due to local homophily, node $u_1$ can be accurately classified, as most of its neighbors share the same label. However, such local interactions are insufficient for classifying nodes $u_2$ and $u_3$. By

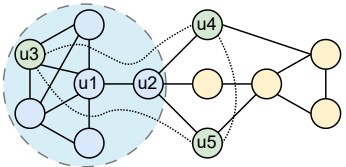

Figure 1: Hierarchical interactions.

leveraging cluster interactions, node $u_2$ can be correctly identified, as it lies within a coherent cluster (the blue region). Furthermore, assuming that nodes labeled green follow a representation distribution distinct from the blue and yellow ones, global interactions-represented by the dotted lines—can be adaptively learned to further enhance the classification of $u_3$. Here, we propose a unified hierarchical mask framework that enables GTs to consistently model such multi-level interactions.

### 3.1 The unified hierarchical mask framework for Graph Transformers

To formalize this framework, we categorize node interactions into three types: (1) N-N: interactions between individual nodes; (2) N-S: interactions between a node and a node set; and (3) S-S: interactions between node sets. For N-N, we directly set $\mathbf{M}_{u,v} = 1$ to indicate a connection from node $u$ to $v$. For N-S and S-S, we treat each node set as a virtual super node $v'$, and extend the node set as $\mathcal{V} = \mathcal{V} \cup \{v'\}$. This allows both N-S and S-S to be equivalently modeled as N-N, enabling a unified and flexible mask design across different interaction levels. Based on this unified framework, we further illustrate how existing GTs implicitly design their attention masks to model hierarchical interactions across local, cluster, and global levels. A summary table is provided in Table 4.

**Local interactions:** Modeling local interactions typically involves capturing information from $K$-hop neighborhoods. GOAT explicitly models N-N interactions between a target node and its $K$-hop

neighbors, which corresponds to the mask $\mathbf{M}^{l1} = \mathbf{A}^K$ [24]. GNN–Transformer hybrid architectures implicitly adopt the mask $\mathbf{M}^{l2} = \mathbf{A}$ through their GNN modules, and aggregating $K$-hop information recursively across layers [43, 11, 44]. Some tokenized GTs aggregate node features at multiple hops as input tokens of a Transformer, effectively applying a set of masks $\{\mathbf{A}^k : 1 \leq k \leq K\}$ [5, 15, 6].

**Cluster interactions:** Several recent GTs have focused on capturing cluster-level interactions by partitioning the graph into disjoint clusters with METIS[22]. We define the partition function $\mathcal{P}(u) = p$ to denote the cluster index assigned to node $u$ and the reverse partition function $\mathcal{P}^{-1}(p) = \{u : \mathbf{C}_{u,p} = 1\}$ to denote the set of nodes belong to cluster $p$. By treating each cluster as a virtual super node, we denote the set of such nodes as $\mathcal{V}^p$. Graph ViT [18] models interactions between clusters (S–S interactions), which corresponds to applying a mask $\mathbf{M}^{c1}$, where $\mathbf{M}^{c1}_{u,v} = 1$ if $u, v \in \mathcal{V}^p$. Cluster-GT [20] models more fine-grained N–S interactions, where each cluster attends to all real nodes. This leads to a mask $\mathbf{M}^{c2}$, defined as $\mathbf{M}^{c2}_{u,v} = 1$ if $u \in \mathcal{V}^p$ and $v \in \mathcal{V}$. To differentiate contributions from different clusters, attention scores are further modulated by the connectivity between clusters and refined via cluster attention implicitly induced by $\mathbf{M}^{c1}$. CoBFormer [44] focuses on N–N interactions within each cluster by implicitly applying a mask $\mathbf{M}^{c3}$, where $\mathbf{M}^{c3}_{u,v} = 1$ if $u, v \in \mathcal{V}$ and $\mathcal{P}(u) = \mathcal{P}(v)$. It additionally applies $\mathbf{M}^{c1}$ to capture inter-cluster interactions.

**Global interactions:** A common strategy for capturing global N–N interactions is to treat the entire graph as fully connected, corresponding to a global mask $\mathbf{M}^{g1} = \mathbf{1}^{N \times N}$ [42, 43, 11]. In contrast, Exphormer [34] approximates global dependencies through an N–S interaction scheme by introducing a set of virtual super nodes $\mathcal{V}^g$, each connected bidirectionally to all real nodes. This yields a global mask $\mathbf{M}^{g2}$, where $\mathbf{M}^{g2}_{u,v} = 1$ if $u \in \mathcal{V}$ and $v \in \mathcal{V}^g$, or $u \in \mathcal{V}^g$ and $v \in \mathcal{V}$.

### 3.2 Theoretical analysis

To investigate the distinct contributions of hierarchical masks to node classification, we develop a theoretical framework based on a class-conditional representation model. Specifically, let $\mathcal{G}$ be a graph with label set $\mathcal{Y}$, and assume a uniform label distribution across nodes. The initial representation of a node with label $c$ is sampled from a $d$-dimensional Gaussian distribution: $\boldsymbol{z} \sim \mathcal{N}(\boldsymbol{\mu}_c, \sigma_c^2 \mathbf{I})$, where $|\boldsymbol{\mu}_c|_2^2 = 1$, and the class prototypes are assumed to be orthogonal, i.e., $\boldsymbol{\mu}_c^\top \boldsymbol{\mu}_{c'} = 0$ for all $c \neq c'$.

Let node $u$ have ground-truth label $c$, and its receptive field be specified by the mask vector $\mathbf{M}_{u,:}$, which contains $k$ non-zero entries. Define $\rho_{c'}$ as the fraction of nodes labeled $c'$ within this receptive field, and $\alpha_{c'}$ as the average attention weight assigned to those nodes. Let $\hat{\boldsymbol{z}}_u$ denote the attention-updated representation of node $u$. We consider a similarity-based classifier that predicts label $c$ correctly if $\hat{\boldsymbol{z}}_u^\top \boldsymbol{\mu}_c \geq \delta_c$, where $\delta_c$ is a decision threshold implicitly learned by models.

**Theorem 3.1.** *The updated representation of node $u$ follows a Gaussian distribution:*

$$\hat{\boldsymbol{z}}_u \sim \mathcal{N} \left( \sum_{i=1}^{|\mathcal{C}|} k\rho_i\alpha_i\boldsymbol{\mu}_i, \ \sum_{i=1}^{|\mathcal{C}|} k\rho_i\alpha_i^2\sigma_i^2 \mathbf{I} \right) \tag{2}$$

*Assume the classifier is well-trained such that $\delta_c - k\rho_c\alpha_c \leq 0$, and the attention weights satisfy the constraint $0 \leq \alpha_{c'} \leq \frac{1}{k} \leq \alpha_c \leq \frac{1}{k\rho_c}$. Then, the probability that node $u$ is correctly classified by a similarity-based classifier is bounded as:*

$$1 - \Phi \left( \frac{\delta_c - k\rho_c\alpha_c}{\sqrt{k\rho_c\alpha_c^2\sigma_c^2 + \frac{1-\rho_c}{k} \cdot \sigma_m^2}} \right) \leq P(\hat{\boldsymbol{z}}_u^\top \boldsymbol{\mu}_c \geq \delta_c) \leq 1 - \Phi \left( \frac{\delta_c - k\rho_c\alpha_c}{\sqrt{k\rho_c\alpha_c^2\sigma_c^2}} \right), \tag{3}$$

*where $\Phi(\cdot)$ denotes the cumulative distribution function (CDF) of the standard normal distribution, and $\sigma_m^2 = \max_{i \neq c} \sigma_i^2$ is the maximum variance among non-target classes.*

*Both the lower and upper bounds are monotonically increasing with respect to $k$, $\rho_c$, and $\alpha_c$, and decreasing with respect to the set of class-wise variances $\{\sigma_i : 1 \leq i \leq |\mathcal{Y}|\}$.*

The proof can be found in Appendix A. Theorem 3.1 shows that the probability of correct classification primarily depends on the factors: $k$, $\rho_c$, $\alpha_c$, and the set of variances $\{\sigma_i : 1 \leq i \leq |\mathcal{Y}|\}$. Since $\alpha_c$ and $\sigma_i$ are affected by the training dynamics and input distribution, respectively, we focus our analysis on $k$ and $\rho_c$, which are determined by the attention mask. Theorem 3.1 demonstrates that larger

values of $k$ and $\rho_c$ lead to higher lower and upper bounds on the probability of correct classification. This gives rise to a fundamental principle for mask construction: An effective attention mask should ensure a sufficiently large receptive field and a high level of label consistency.

Building on this theorem, we further analyze the applicability of hierarchical masks derived from existing GTs under several representative scenarios: 1) For nodes with strong local homophily, using local masks (e.g., $\mathbf{M}^{l1}$, $\mathbf{M}^{l2}$) is effective due to their typically large $\rho_c$. 2) For nodes near cluster boundaries, local homophily weakens as some neighbors belong to other clusters. In such cases, cluster masks (e.g., $\mathbf{M}^{c3}$) can yield higher $\rho_c$ and a larger $k$, potentially leading to improved classification performance, assuming the partitioning is accurate. 3) For heterophilic nodes with minority labels in their cluster, local or cluster masks may result in very small $\rho_c$, even below $\frac{1}{|\mathcal{Y}|}$. Here, the global mask $\mathbf{M}^{g1}$ is preferable, as it ensures $\rho_c = \frac{1}{|\mathcal{Y}|}$ and provides a large $k = N$. By contrast, the use of $\mathbf{M}^{g2}$ may be less effective, as the global virtual nodes lack explicit label semantics. 4) Some prior works approximate global interactions through inter-cluster attention (e.g., via $\mathbf{M}^{c1}$). However, as shown in Equation 2, cluster-level representations are dominated by the majority label. During inter-cluster aggregation, this bias is further amplified, increasing the risk of misclassifying minority-label nodes as the dominant class. 5) For nodes belonging to classes with well-defined representation distributions (i.e., small variance $\sigma_c$), the attention weights $\alpha_c$ can be effectively learned to approximate $\frac{1}{k\rho_c}$, leading to a higher probability of correct classification regardless of the specific mask employed.

The above analysis suggests that no single mask consistently satisfies the proposed principle across all scenarios. However, hierarchical masks at different levels offer complementary strengths in node classification, and their integration provides a natural means of aligning with the principle.

Table 1: Accuracy comparison of individual masks, ensemble strategies, and the oracle case.

| Dataset | Local | Cluster | Global | Mean | Max | Oracle |
|---|---|---|---|---|---|---|
| **Cora** | **87.71**$_{\pm 1.30}$ | 82.10$_{\pm 1.53}$ | 73.03$_{\pm 1.53}$ | 86.41$_{\pm 1.72}$ | 86.82$_{\pm 2.32}$ | _93.41$_{\pm 1.38}$_ |
| **Citeseer** | **77.02**$_{\pm 2.10}$ | 71.43$_{\pm 2.39}$ | 72.94$_{\pm 2.10}$ | 76.16$_{\pm 2.10}$ | 75.73$_{\pm 2.08}$ | _84.32$_{\pm 2.10}$_ |
| **Pubmed** | **89.77**$_{\pm 0.46}$ | 87.96$_{\pm 0.35}$ | 87.51$_{\pm 0.42}$ | 89.31$_{\pm 0.28}$ | 89.19$_{\pm 0.35}$ | _93.68$_{\pm 0.25}$_ |
| **Photo** | 94.25$_{\pm 0.46}$ | 94.26$_{\pm 1.06}$ | 85.61$_{\pm 4.26}$ | **95.14**$_{\pm 0.52}$ | 94.77$_{\pm 0.74}$ | _97.50$_{\pm 0.40}$_ |
| **Computer** | **91.57**$_{\pm 0.63}$ | 89.33$_{\pm 0.79}$ | 82.78$_{\pm 1.21}$ | 91.41$_{\pm 0.92}$ | 90.45$_{\pm 0.73}$ | _95.59$_{\pm 0.37}$_ |
| **Squirrel** | 38.67$_{\pm 1.72}$ | 38.03$_{\pm 1.10}$ | 38.64$_{\pm 1.68}$ | 38.53$_{\pm 1.16}$ | **38.78**$_{\pm 1.34}$ | _53.72$_{\pm 1.30}$_ |
| **Chameleon** | 41.97$_{\pm 3.90}$ | 42.24$_{\pm 3.35}$ | **43.50**$_{\pm 2.97}$ | 42.96$_{\pm 4.29}$ | 42.60$_{\pm 4.58}$ | _64.57$_{\pm 4.05}$_ |

### 3.3 Experimental analysis

To further investigate whether combining masks across multiple interaction levels yields benefits in real-world scenarios, we construct three GTs trained separately with the local mask $\mathbf{M}^{l2}$, cluster mask $\mathbf{M}^{c3}$, and global mask $\mathbf{M}^{g1}$ on seven node classification datasets. Then we apply three ensemble strategies to integrate their outputs: Mean: averaging the predicted probabilities; Max: selecting the maximum predicted probability; and Oracle: an idealized upper bound where the best prediction is always selected. The performance of individual models and ensemble methods is reported in Table 1, with three key observations: 1) The strong performance of Oracle indicates that properly integrating complementary information from multiple interaction levels can yield substantial performance gains. 2) Naive ensemble methods, such as Mean and Max, underperform the best single-mask model on 5 out of 7 datasets, revealing a key challenge for Graph Transformers: *How to effectively integrate multi-level interaction information?*

Moreover, our experiments reveal a significant efficiency issue: even a 2-layer Transformer with 2 heads and a single mask, consumes 21 GB of GPU memory on the PubMed. Although several efficient Transformer variants—such as kernel-based linear attention methods [8, 35] and FlashAttention [9]—have been proposed to alleviate the $O(N^2)$ complexity, their applicability to graphs remains limited due to the irregular and diverse patterns of graph masks [39]. This leads to another key challenge: *How to efficiently implement GTs with irregular masks on large-scale graphs?*

## 4 Method

In this section, we propose M$^3$Dphormer, a novel **M**ixture-of-Experts based Gra**ph** Transf**ormer** with **M**ulti-Level **M**asking and **D**ual Attention Computation. The overall framework is illustrated

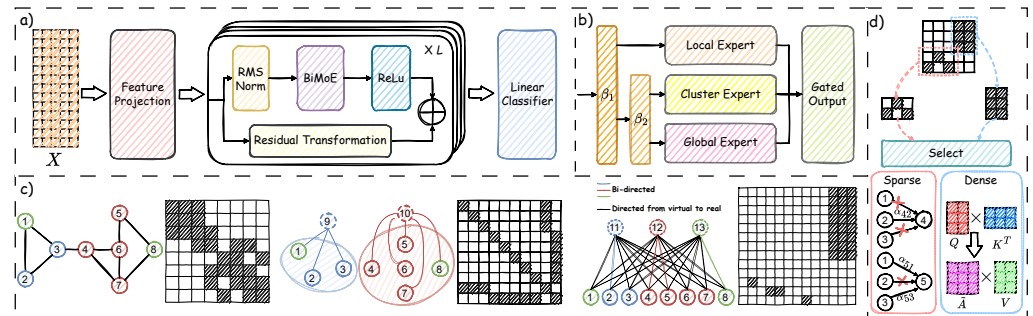

Figure 2: Overview of the M³Dphormer with Pre-RMSNorm [48] and ReLU[1]: a) The overall network architecture. b) The bi-level expert routing mechanism. c) The theorem-guided hierarchical mask design strategy. Self-connections are added for nodes in local and cluster masks. Nodes 2, 4, 5 and 8 are included in the training set. d) The dual attention computation scheme.

in Figure 2. Guided by theoretical analysis, we first design three hierarchical attention masks to comprehensively model multi-level interactions (part **c**). To adaptively integrate information across these interaction levels, we introduce a bi-level attention expert routing mechanism, where each expert is a MHA module associated with a specific mask (part **b**). Additionally, a dual attention computation scheme is incorporated to ensure computational efficiency (part **d**).

## 4.1 Overall architecture of M³Dphormer

We first present the overall architecture of M³Dphormer. The initial representation is given by $\mathbf{H}^0 = \mathbf{X}\mathbf{W}_{in}$, where $\mathbf{W}_{in} \in \mathbb{R}^{d_{in} \times d}$ is a learnable linear projection. The model then applies $L$ stacked M³Dphormer layers, with the computation in the $l$-th layer defined as:

$$\mathbf{H}^l = \text{ACT}\left(\text{BiMoE}^l\left(\text{Norm}^l(\mathbf{H}^{l-1}), \boldsymbol{\mathcal{M}}\right)\right) + \mathbf{H}^{l-1}\mathbf{W}_{res}^l, \tag{4}$$

where $\mathbf{H}^{l-1}$ is the input from the previous layer, $\mathbf{W}_{res}^l$ is the residual projection matrix following [26], $\text{Norm}^l(\cdot)$ is the normalization function, and $\text{ACT}(\cdot)$ is the activation function. The bi-level attention expert routing mechanism $\text{BiMoE}^l(\cdot)$ integrates information from multiple interaction levels based on the hierarchical mask set $\boldsymbol{\mathcal{M}}$. After $L$ layers, a linear classifier is applied to obtain the final prediction: $\hat{\mathbf{Y}} = \mathbf{H}^L\mathbf{W}_{cls}$, where $\mathbf{W}_{cls}$ is the classifier weight matrix. The model is optimized using cross-entropy loss computed over both the training nodes $\mathcal{V}_{\text{train}}$ and the label-specific global virtual nodes $\mathcal{V}^g$, which are introduced in the following subsection.

## 4.2 Theorem-guided hierarchical mask design strategy

To enable comprehensive modeling of multi-level node interactions, we propose a theoretically grounded design strategy for the hierarchical mask set $\boldsymbol{\mathcal{M}} = \{\mathbf{M}^{l2}, \mathbf{M}^{c4}, \mathbf{M}^{g3}\}$.

**Local mask design:** We adopt $\mathbf{M}^{l2} = \mathbf{A}$ as the local mask due to its several advantages over $\mathbf{M}^{l1} = \mathbf{A}^K$: 1) As observed in CoBFormer[44], the homophily ratio $\rho_c$ tends to decline rapidly with increasing hop size $K$. A lower $\rho_c$ may lead to reduced classification probability, as indicated by Theorem 3.1. 2) $\mathbf{M}^{l1}$ ignores distance information between the target node and its $K$-hop neighbors, thus requiring explicit distance-aware position encoding[24]. In contrast, $\mathbf{M}^{l2}$ can capture such distance implicitly through recursive aggregation across layers. 3) $\mathbf{M}^{l2}$ is sparser than $\mathbf{M}^{l1}$, enabling more efficient computation under the dual attention scheme introduced later.

**Cluster mask design:** We first partition the graph into $P$ disjoint clusters using METIS[22], and introduce a set of cluster-level virtual nodes $\mathcal{V}^p = \{N + i : 0 \le i < P\}$. We then define a new cluster mask $\mathbf{M}^{c4}$, where $\mathbf{M}^{c4}_{u,v} = 1$ if either (i) $u \in \mathcal{V}$ and $v \in \{u, N + \mathcal{P}(u)\}$, or (ii) $u \in \mathcal{V}^p$ and $v \in \mathcal{P}^{-1}(u - N)$. The functions $\mathcal{P}(\cdot)$ and $\mathcal{P}^{-1}(\cdot)$ denote the partition and reverse-partition function defined in Section 3.1. This formulation restricts attention to node–cluster pairs within the same partition. The feature of each virtual node in $\mathcal{V}^p$ is computed by averaging the features of the real nodes in its corresponding cluster. Compared to $\mathbf{M}^{c3}$, the proposed $\mathbf{M}^{c4}$ significantly reduces the

non-zero ratio from $\frac{1}{P}$ to $\frac{3N}{(N+P)^2}$ since $P \ll N$, making it more efficient. Moreover, as demonstrated in Proposition 4.1, the interactions captured by $\mathbf{M}^{c3}$ can be effectively approximated using $\mathbf{M}^{c4}$.

**Proposition 4.1.** *Cluster interactions modeled by a single Graph Transformer layer using $\mathbf{M}^{c3}$ can be equivalently modeled by two consecutive layers using $\mathbf{M}^{c4}$.*

The proof is given in Appendix A.4. Although this requires an additional layer, the reduction in sparsity still leads to substantial computational savings by our dual attention computation scheme.

**Global mask design:** We introduce a new global mask $\mathbf{M}^{g3}$, which extends $\mathbf{M}^{g2}$ by explicitly incorporating label semantics. Specifically, we add $|\mathcal{Y}|$ global virtual nodes, indexed as $\mathcal{V}^g = \{N + P + i : 0 \leq i < |\mathcal{Y}|\}$, each associated with a distinct class label. The mask $\mathbf{M}^{g3}_{u,v}$ is set to 1 if either: (i) $u \in \mathcal{V}$ and $v \in \mathcal{V}^g$, or (ii) $u \in \mathcal{V}^g$ and $v \in \{t \in \mathcal{V}_{\text{train}} : \mathbf{y}_t = \mathbf{y}_u\}$. This structure enables each real node to attend to all global nodes, while each global node aggregates information only from training nodes with a specific label. Similar to $\mathcal{V}^p$, the features of nodes in $\mathcal{V}^g$ are obtained by averaging the features of training nodes with the corresponding label. Let $g_c$ denote the global virtual node for class $c$, and $n_c$ be the number of training nodes with label $c$. By Equation 2, the updated representation satisfies: $\hat{\mathbf{z}}_{g_c} \sim \mathcal{N}(\boldsymbol{\mu}_c, \frac{\sigma^2}{n_c}\mathbf{I})$, since $\rho_c = 1$, $\rho_{c'} = 0$ for $c' \neq c$, and $\alpha_i = \frac{1}{n_c}$. The reduced variance $\frac{\sigma^2}{n_c}$ indicates that it is more concentrated around the class mean. According to Theorem 3.1, this leads to improved bounds for the probability of correct classification.

### 4.3 Bi-level attention expert routing mechanism

To adaptively integrate information from different interaction levels, we propose the bi-level attention expert routing mechanism—the core component of $M^3$Dphormer. Each expert corresponds to an MHA module equipped with a specific attention mask. Motivated by the observation in Table 1 that the local mask yields the best performance in most cases, we prioritize the local expert at the first routing level. The second level then refines the selection among the cluster and global experts. The bi-level routing mechanism is formally defined as:

$$\text{BiMoE}(\mathbf{H}, \boldsymbol{\mathcal{M}}) = \mathbf{g}_1 \cdot \text{MHA}^D(\mathbf{H}, \mathbf{M}^{l2}) + \mathbf{g}_2 \cdot \text{MHA}^D(\mathbf{H}, \mathbf{M}^{c4}) + \mathbf{g}_3 \cdot \text{MHA}^D(\mathbf{H}, \mathbf{M}^{g3})$$
$$\mathbf{g}_1 = \boldsymbol{\beta}_1, \quad \mathbf{g}_2 = (1 - \boldsymbol{\beta}_1) \cdot \boldsymbol{\beta}_2, \quad \mathbf{g}_3 = (1 - \boldsymbol{\beta}_1) \cdot (1 - \boldsymbol{\beta}_2) \tag{5}$$
$$\boldsymbol{\beta}_1 = \text{Sigmoid}(\mathbf{H}\mathbf{W}_G^1), \quad \boldsymbol{\beta}_2 = \text{Sigmoid}(\mathbf{H}\mathbf{W}_G^2)$$

Here, $\boldsymbol{\mathcal{M}}$ denotes the hierarchical mask set. $\text{MHA}^D(\cdot, \cdot)$ denotes our proposed dual attention computation scheme. $\mathbf{W}_G^1, \mathbf{W}_G^2 \in \mathbb{R}^{d \times 1}$ are learnable gating parameters for the first- and second-level expert selection. The sigmoid function constrains the gating values $\boldsymbol{\beta}_1$ and $\boldsymbol{\beta}_2$ within $[0, 1]$. To emphasize the empirical importance of local interactions, both $\mathbf{W}_G^1$ and $\mathbf{W}_G^2$ are initialized as zero vectors, yielding initial routing weights of $[0.5, 0.25, 0.25]$ for each node, which prioritizes local attention in the early training stage. Inspired by the observation in Table 1 that all interaction levels contribute significantly to the classification, we aggregate outputs from all experts using the learned routing weights $\mathbf{g}_1$, $\mathbf{g}_2$, and $\mathbf{g}_3$, without applying top-$k$ selection.

### 4.4 Dual attention computation scheme

Finally, we introduce the dual attention computation scheme to enhance computational efficiency. While the irregularity of graph masks hinders the application of efficient attention variants[8, 9, 39], their inherent sparsity enables a new optimization route—*sparse attention computation*[34]. Unlike standard dense attention (Equation 1), which constructs the full attention matrix $\hat{\mathbf{A}}$ before applying the binary mask $\mathbf{M}$, sparse attention computes attention scores only for valid node pairs $(u, v)$ where $\mathbf{M}_{u,v} = 1$. A detailed analysis of the sparse attention implementation in Algorithm 2 reveals a space complexity of $O(6mHd_h)$, where $m$ is the number of non-zero entries in $\mathbf{M}$, $H$ is the number of attention heads, and $d_h$ is hidden dimension of each head. Although this method significantly reduces the complexity from $O(N^2)$ to $O(m)$, the constant factor $6Hd_h$ remains non-negligible in practice.

To further improve efficiency, we propose a dual attention computation scheme that dynamically switches between dense and sparse computation based on the local sparsity of the attention mask. Specifically, we partition the attention mask $\mathbf{M}$ into $K$ disjoint regions $\mathcal{R} = \{\mathcal{R}_i\}_{i=1}^K$, each $\mathcal{R}_i$ is defined by a query set $\mathcal{Q}_i$ and the corresponding key set $\mathcal{K}_i = \cup_{u \in \mathcal{Q}_i}\{v : \mathbf{M}_{u,v} = 1\}$. For each region, the optimal computation mode is selected according to Proposition 4.2.

Table 2: Node classification results ($\%_{\pm\sigma}$). ROC-AUC for Minesweeper; accuracy for the rest.

| Models | Cora | Citeseer | Pubmed | Computer | Photo | Squirrel | Chameleon | Minesweeper | Arxiv |
|---|---|---|---|---|---|---|---|---|---|
| GCN | $86.53_{\pm1.61}$ | $75.97_{\pm1.93}$ | $88.51_{\pm0.28}$ | $89.83_{\pm0.64}$ | $93.07_{\pm0.48}$ | $42.19_{\pm2.10}$ | $42.87_{\pm2.78}$ | $93.47_{\pm0.47}$ | $72.55_{\pm0.21}$ |
| GAT | $86.53_{\pm1.27}$ | $74.31_{\pm1.25}$ | $87.42_{\pm0.43}$ | $90.45_{\pm0.87}$ | $93.79_{\pm0.28}$ | $36.59_{\pm1.88}$ | $41.52_{\pm4.78}$ | $93.25_{\pm0.42}$ | $72.10_{\pm0.33}$ |
| SAGE | $87.62_{\pm1.73}$ | $74.79_{\pm1.59}$ | $89.20_{\pm0.53}$ | $90.14_{\pm0.73}$ | $94.27_{\pm0.64}$ | $36.16_{\pm1.46}$ | $42.06_{\pm3.01}$ | $93.64_{\pm0.39}$ | $72.32_{\pm0.13}$ |
| GCN* | $87.74_{\pm1.66}$ | $76.83_{\pm1.94}$ | $89.48_{\pm0.56}$ | $91.70_{\pm0.53}$ | $95.10_{\pm0.60}$ | $42.59_{\pm2.14}$ | $43.77_{\pm2.47}$ | $97.39_{\pm0.30}$ | $73.18_{\pm0.21}$ |
| GAT* | $87.59_{\pm1.33}$ | $76.42_{\pm1.81}$ | $89.12_{\pm0.47}$ | $91.70_{\pm0.73}$ | $95.73_{\pm0.63}$ | $38.38_{\pm1.25}$ | $42.69_{\pm5.11}$ | $97.35_{\pm1.12}$ | $72.67_{\pm0.14}$ |
| SAGE* | $87.71_{\pm1.91}$ | $75.99_{\pm1.21}$ | $89.49_{\pm0.29}$ | $91.52_{\pm0.60}$ | $95.37_{\pm0.89}$ | $39.28_{\pm2.90}$ | $43.23_{\pm2.78}$ | $97.39_{\pm0.80}$ | $72.75_{\pm0.14}$ |
| GPRGNN | $88.21_{\pm1.29}$ | $77.02_{\pm1.81}$ | $88.49_{\pm0.44}$ | $90.70_{\pm0.53}$ | $94.80_{\pm0.37}$ | $36.80_{\pm1.71}$ | $41.26_{\pm4.22}$ | $89.05_{\pm0.43}$ | $68.44_{\pm0.21}$ |
| FAGCN | $88.36_{\pm1.50}$ | $76.78_{\pm1.29}$ | $89.31_{\pm0.58}$ | $90.07_{\pm0.72}$ | $95.23_{\pm0.38}$ | $40.90_{\pm2.04}$ | $42.78_{\pm3.51}$ | $89.95_{\pm0.62}$ | $66.83_{\pm0.20}$ |
| NAGphormer | $87.68_{\pm1.80}$ | $76.21_{\pm2.72}$ | $89.35_{\pm0.20}$ | $91.22_{\pm0.65}$ | $95.12_{\pm0.36}$ | $38.67_{\pm1.47}$ | $41.61_{\pm2.64}$ | $90.08_{\pm0.46}$ | $71.38_{\pm0.20}$ |
| Exphormer | $87.03_{\pm1.70}$ | $76.18_{\pm1.50}$ | $88.55_{\pm0.47}$ | $90.76_{\pm0.80}$ | $95.19_{\pm0.49}$ | $37.20_{\pm2.56}$ | $40.27_{\pm1.63}$ | $95.32_{\pm0.93}$ | $72.24_{\pm0.21}$ |
| SGFormer | $87.86_{\pm1.12}$ | $75.85_{\pm2.04}$ | $88.75_{\pm0.41}$ | $91.52_{\pm0.68}$ | $95.10_{\pm0.32}$ | $38.60_{\pm0.95}$ | $43.77_{\pm3.02}$ | $91.59_{\pm0.28}$ | $72.44_{\pm0.28}$ |
| CoBFormer | $88.15_{\pm1.47}$ | $77.05_{\pm1.69}$ | $88.50_{\pm0.59}$ | $91.64_{\pm0.41}$ | $95.58_{\pm0.55}$ | $39.03_{\pm1.35}$ | $43.50_{\pm1.35}$ | $95.63_{\pm0.52}$ | $73.17_{\pm0.18}$ |
| PolyNormer | $87.83_{\pm1.94}$ | $76.93_{\pm2.16}$ | $89.48_{\pm0.43}$ | $91.85_{\pm0.57}$ | $95.44_{\pm0.71}$ | $39.32_{\pm1.45}$ | $44.30_{\pm2.04}$ | $96.98_{\pm0.46}$ | $73.27_{\pm0.38}$ |
| Mowst | $87.92_{\pm1.17}$ | $76.52_{\pm1.41}$ | $88.71_{\pm0.25}$ | $91.32_{\pm0.50}$ | $93.70_{\pm0.32}$ | $41.72_{\pm2.33}$ | $44.30_{\pm2.57}$ | $93.00_{\pm0.68}$ | $73.03_{\pm0.29}$ |
| GCN-MoE | $86.17_{\pm1.11}$ | $75.20_{\pm1.70}$ | $88.80_{\pm0.24}$ | $88.75_{\pm0.58}$ | $93.19_{\pm0.32}$ | $43.02_{\pm2.14}$ | $44.57_{\pm2.00}$ | $92.63_{\pm0.40}$ | $73.16_{\pm0.21}$ |
| $M^3$Dphormer | $\mathbf{88.48_{\pm1.94}}$ | $\mathbf{77.53_{\pm1.56}}$ | $\mathbf{89.96_{\pm0.49}}$ | $\mathbf{92.09_{\pm0.46}}$ | $\mathbf{95.91_{\pm0.68}}$ | $\mathbf{44.34_{\pm1.94}}$ | $\mathbf{47.09_{\pm4.05}}$ | $\mathbf{98.27_{\pm0.20}}$ | $\mathbf{73.54_{\pm0.30}}$ |

**Proposition 4.2.** *Let $\kappa_{\mathcal{R}_i}$ denote the non-zero rate within region $\mathcal{R}_i$. The sparse attention scheme is more efficient than the dense scheme when $\kappa_{R_i} < \frac{1}{3d_h}$.*

The proof is provided in Appendix A.5. This result guides the selection of sparse computation when local sparsity is high and dense computation when the region becomes sufficiently dense. Based on this guidance, we formulate the dual attention computation scheme as:

$$\mathrm{MHA}^D(\mathbf{H}, \mathbf{M}) = \mathrm{Comb}_i^K\left(\mathrm{SelectMode}\left(\mathcal{R}_i\right)\left(\mathbf{H}, \mathcal{R}_i\right)\right) \tag{6}$$

Here, SelectMode$(\mathcal{R}_i)$ denotes the selection between sparse and dense computation modes for region $\mathcal{R}_i$, and $\mathrm{Comb}_i^K(\cdot)$ aggregates the attention outputs from all $K$ partitioned regions.

## 5 Experiments

**Experiment setups.** We evaluate $M^3$Dphormer on nine datasets, including six homophilic graphs (Cora, CiteSeer, Pubmed [45], Computer, Photo [33], and Ogbn-Arxiv [19]) and three heterophilic graphs (Squirrel, Chameleon, and Minesweeper [31]). Dataset statistics and splitting protocols are detailed in Appendix D. We select 15 baselines spanning five categories: 1) *Classic GNNs*: GCN [23], GAT [38], and GraphSAGE [16]. 2) *Enhanced Classic GNNs*: GCN*, GAT*, and SAGE*. 3) *Advanced GNNs*: GPRGNN[7] and FAGCN [3]. 4) *SOTA Graph Transformers*: NAGphormer [5], Exphormer [34], SGFormer [43], CoBFormer [44], and PolyNormer [11]; 5) *MoE-based GNNs*: Mowst [47] and GCN-MoE [40]. Descriptions of the baselines are provided in Appendix E, and implementation details can be found in Appendix F.

**Node classification results.** Table 2 presents the node classification results. Key observations include: 1) $M^3$Dphormer consistently outperforms all baselines across 9 datasets, highlighting its superior interaction modeling capacity. 2) Compared to traditional and MoE-based GNNs, $M^3$Dphormer demonstrates clear advantages by comprehensively capturing hierarchical interactions. 3) Compared to GT baselines, $M^3$Dphormer shows notable improvements, verifying both the benefit of comprehensive interaction modeling and the effectiveness of the bi-level attention expert routing mechanism in adaptively integrating multi-level information.

**Ablation studies.** We perform ablation studies to evaluate $M^3$Dphormer in terms of effectiveness and efficiency. Firstly, we construct five ablated variants by: 1) Removing individual experts; 2) Disabling

Table 3: Node classification results of various M³Dphormer variants

| | Cora | Citeseer | Pubmed | Computer | Photo | Squirrel | Chameleon | Minesweeper | Ogbn-Arxiv |
|---|---|---|---|---|---|---|---|---|---|
| **Full Model** | $\mathbf{88.48}_{\pm 1.94}$ | $\mathbf{77.53}_{\pm 1.56}$ | $\mathbf{89.96}_{\pm 0.49}$ | $\mathbf{92.09}_{\pm 0.46}$ | $\mathbf{95.91}_{\pm 0.68}$ | $\mathbf{44.34}_{\pm 1.94}$ | $\mathbf{47.09}_{\pm 4.05}$ | $\mathbf{98.27}_{\pm 0.20}$ | $\mathbf{73.54}_{\pm 0.30}$ |
| W/O Local | $82.84_{\pm 2.15}$ | $74.09_{\pm 1.46}$ | $88.75_{\pm 0.48}$ | $88.99_{\pm 0.94}$ | $93.83_{\pm 0.44}$ | $39.61_{\pm 1.51}$ | $42.60_{\pm 4.41}$ | $57.55_{\pm 0.78}$ | $67.24_{\pm 0.23}$ |
| W/O Cluster | $87.83_{\pm 1.83}$ | $76.73_{\pm 2.00}$ | $89.69_{\pm 0.46}$ | $91.59_{\pm 0.70}$ | $95.22_{\pm 0.53}$ | $42.48_{\pm 2.13}$ | $44.93_{\pm 3.21}$ | $98.03_{\pm 0.41}$ | $73.40_{\pm 0.23}$ |
| W/O Global | $87.95_{\pm 2.03}$ | $76.33_{\pm 1.84}$ | $89.76_{\pm 0.34}$ | $91.89_{\pm 0.50}$ | $95.62_{\pm 0.59}$ | $41.58_{\pm 1.60}$ | $45.47_{\pm 5.04}$ | $98.02_{\pm 0.24}$ | $73.41_{\pm 0.11}$ |
| W/O Route | $87.65_{\pm 2.39}$ | $76.25_{\pm 0.91}$ | $89.48_{\pm 0.46}$ | $91.76_{\pm 0.71}$ | $95.59_{\pm 0.87}$ | $42.05_{\pm 1.21}$ | $43.95_{\pm 1.98}$ | $97.72_{\pm 0.17}$ | $73.31_{\pm 0.19}$ |
| W/O Bi-Level | $87.74_{\pm 2.21}$ | $77.12_{\pm 1.59}$ | $89.79_{\pm 0.25}$ | $91.68_{\pm 0.37}$ | $95.70_{\pm 0.50}$ | $42.41_{\pm 1.96}$ | $44.39_{\pm 4.44}$ | $97.84_{\pm 0.34}$ | $73.50_{\pm 0.17}$ |

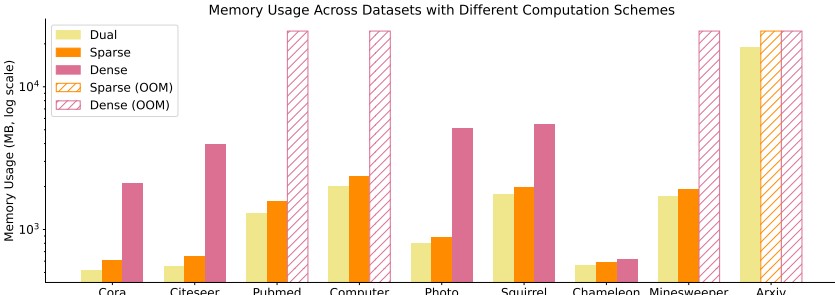

Figure 3: Memory usage of different schemes.

the attention expert routing mechanism; 3) Replacing the bi-level attention routing mechanism with a single-level gating scheme. Results in Table 3 show that: 1) Removing any individual expert consistently degrades performance, underscoring the necessity of comprehensively modeling hierarchical interactions. 2) Disabling the routing mechanism leads to substantial degradation, suggesting that simple aggregation is insufficient for effectively integrating multi-level interactions. 3) The performance gap between the single-level routing variant and M³Dphormer demonstrates the advantage of the proposed bi-level attention expert routing mechanism. Then, we report the GPU memory usage of M³Dphormer and its two variants employing sparse and dense computation schemes in Figure 3. As shown, the dense scheme incurs the highest memory consumption and leads to out-of-memory (OOM) errors on four datasets. While the sparse scheme substantially reduces memory usage, it still fails to run on Ogbn-Arxiv. In contrast, our dual attention scheme achieves superior memory efficiency and successfully scales to all evaluated graphs.

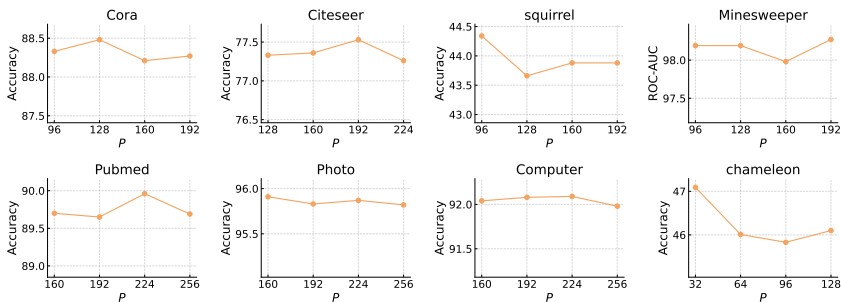

Figure 4: Test accuracy vs. $P$

**Parameter analysis.** The only key hyperparameter in M³Dphormer is the number of clusters $P$, which affects the quality of the cluster mask $\mathbf{M}^{c4}$. For each dataset, we select an appropriate range of $P$ values according to its size. Figure 4 presents the model's performance under varying $P$. Overall, M³Dphormer demonstrates strong robustness to the choice of $P$ on most datasets. An exception is Chameleon—a small graph with only 890 nodes—where variations in $P$ substantially impact the quality of partitioning, resulting in more significant performance fluctuations.

**Visualization.** We plot the accuracy and loss curves of M³Dphormer and GCN*[23, 26] in Figure 5. Across the training, validation, and test sets, M³Dphormer consistently achieves faster convergence and higher accuracy than GCN* during the training process, demonstrating the effectiveness of our method. A similar trend is also observed in the comparison with PolyNormer [11] in Appendix G.4. A visualization of the learned gate weights is provided in Appendix G.5.

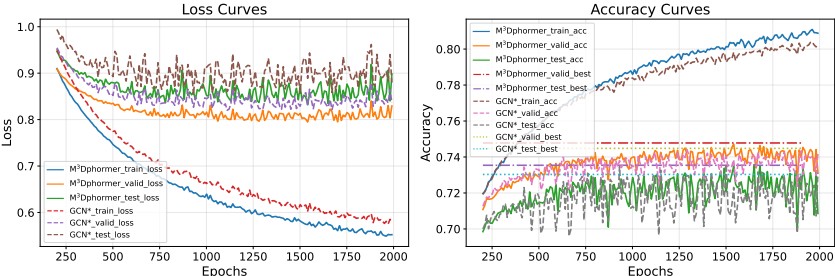

Figure 5: Comparison of accuracy and loss curves between M³DFormer and GCN* on Ogbn-Arxiv.

# 6 Related Work

**Graph neural networks.** Classic GNNs, such as GCN [23], SAGE [16], and GAT [38], rely on message-passing mechanisms that recursively aggregate information from local neighbors at each layer. To move beyond the widely adopted homophily assumption—that nodes of the same class are more likely to be connected [28]—advanced GNNs such as GPRGNN [7] and FAGCN [3] have been proposed to improve performance on heterophilic graphs. More recently, a benchmark study demonstrated that carefully tuning the hyperparameters of classic GNNs and enhancing them with advanced training techniques—such as residual connections [17] and normalization methods [48, 2, 21]—can lead to substantial improvements in node classification performance. While GNNs have proven effective in many scenarios, they still suffer from a fundamental limitation: message passing primarily captures local interactions, often neglecting informative signals from broader, long-range dependencies.

**Graph transformers.** Recently, Graph Transformers have emerged as a promising paradigm for graph representation learning. By leveraging the multi-head self-attention mechanism originally introduced in Transformer [37], they aim to adaptively model diverse and complex interactions from a broader perspective. A primary line of research treats the entire graph as fully connected and computes attention scores between all node pairs [46, 32, 43, 11]. However, a recent study has revealed the over-globalizing problem in such methods, which may lead to a significant decrease in performance [44]. Another line of work constructs a token sequence for each node based on the graph structure and feeds these sequences into a Transformer to learn node representations [49, 5, 15, 6, 41]. These methods often rely on expert-designed tokenization strategies, which tend to capture local information while overlooking larger-scale interactions. In addition, several recent approaches focus on modeling cluster-level interactions, which have shown promising results in capturing mid-level structural patterns for graph representation learning [18, 20, 44].

# 7 Conclusion

In this paper, we propose a unified hierarchical mask framework for Graph Transformers. A fundamental design principle and two core challenges are identified under this framework. We then introduce M³Dphormer, a novel Mixture-of-Experts based Graph Transformer with Multi-Level Masking and Dual Attention Computation, designed to efficiently, comprehensively and adaptively capture hierarchical interactions. Extensive experiments demonstrate its effectiveness.

**Limitations and broader impacts.** In this paper, we focus our theoretical and empirical analyses on the node classification task under the proposed unified hierarchical mask framework, as it is a fundamental and extensively studied problem in the graph learning community. Extending our theoretical insights to graph-level and edge-level tasks represents a promising direction for future work. Apart from this, we do not expect any direct negative societal impacts.

## Acknowledgments and Disclosure of Funding

This work is supported in part by the National Natural Science Foundation of China (No. U20B2045, 62322203, 62172052, 62192784, U22B2038).

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

# A  Theorem Proofs

We begin by decomposing Theorem 3.1 into two theorems and one proposition, and then prove them individually.

**Theorem A.1.** *The updated representation of node $u$ follows a Gaussian distribution:*

$$\hat{\boldsymbol{z}}_u \sim \mathcal{N}\left(\sum_{i=1}^{|\mathcal{C}|} k\rho_i\alpha_i\boldsymbol{\mu}_i, \; \sum_{i=1}^{|\mathcal{C}|} k\rho_i\alpha_i^2\sigma_i^2\mathbf{I}\right) \tag{7}$$

*Accordingly, the probability that node $u$ is correctly classified by a similarity-based classifier is:*

$$P(\hat{\boldsymbol{z}}_u^\top\boldsymbol{\mu}_c \geq \delta_c) = 1 - \Phi\left(\frac{\delta_c - k\rho_c\alpha_c}{\sqrt{\sum_{i=1}^{|\mathcal{C}|} k\rho_i\alpha_i^2\sigma_i^2}}\right) \tag{8}$$

*where $\Phi(\cdot)$ denotes the cumulative distribution function (CDF) of the standard normal distribution.*

**Theorem A.2.** *Assuming the classifier is well trained, i.e., $\delta_c - k\rho_c\alpha_c \leq 0$, and the attention weights satisfy $0 \leq \alpha_{c'} \leq \frac{1}{k} \leq \alpha_c \leq \frac{1}{k\rho_c}$, then the classification probability is bounded as:*

$$1 - \Phi\left(\frac{\delta_c - k\rho_c\alpha_c}{\sqrt{k\rho_c\alpha_c^2\sigma_c^2 + \frac{1-\rho_c}{k}\cdot\sigma_m^2}}\right) \leq P(\hat{\boldsymbol{z}}_u^\top\boldsymbol{\mu}_c \geq \delta_c) \leq 1 - \Phi\left(\frac{\delta_c - k\rho_c\alpha_c}{\sqrt{k\rho_c\alpha_c^2\sigma_c^2}}\right), \tag{9}$$

*where $\sigma_m^2 = \max_{c'\neq c}\sigma_{c'}^2$ is the largest representation variance among non-target classes.*

**Proposition A.3.** *In Equation 9, both the lower and upper bounds are monotonically increasing with respect to $k$, $\rho_c$, and $\alpha_c$, and decreasing with respect to the set of variances $\{\sigma_i : 1 \leq i \leq |\mathcal{Y}|\}$.*

## A.1  Proof of Theorem A.1

*Proof.* Let node $u$ have ground-truth label $c$, and denote its receptive field by the mask vector $\mathbf{M}_{u,:}$, which includes $k$ nodes in total. Let $\rho_i$ be the proportion of class-$i$ nodes in the receptive field, and let $\alpha_i$ denote the average attention weight assigned to these nodes.

The updated representation of node $u$ is given by:

$$\hat{\boldsymbol{z}}_u = \sum_{i=1}^{k} \alpha_i\boldsymbol{z}_i \tag{10}$$

Each node representation $\boldsymbol{z}_i$ with class label $c'$ is generated using the reparameterization trick:

$$\boldsymbol{z}_i^{(c')} = \boldsymbol{\mu}_{c'} + \sigma_{c'}\cdot\boldsymbol{\xi}_i, \quad \text{where } \boldsymbol{\xi}_i \sim \mathcal{N}(\mathbf{0}, \mathbf{I}) \tag{11}$$

Regrouping the neighbors by class, we rewrite the update as:

$$\hat{\boldsymbol{z}}_u = \sum_{i=1}^{|\mathcal{Y}|}\sum_{j=1}^{k\rho_i} \alpha_i\boldsymbol{z}_j^{(i)} = \sum_{i=1}^{|\mathcal{Y}|} k\rho_i\alpha_i\boldsymbol{\mu}_i + \sum_{i=1}^{|\mathcal{Y}|} \sigma_i\alpha_i\sum_{j=1}^{k\rho_i}\boldsymbol{\xi}_j \tag{12}$$

Since the sum of i.i.d. standard Gaussian vectors satisfies:

$$\sum_{j=1}^{k\rho_i}\boldsymbol{\xi}_j \sim \mathcal{N}(\mathbf{0}, k\rho_i\mathbf{I}), \tag{13}$$

we have:

$$\sigma_i\alpha_i\sum_{j=1}^{k\rho_i}\boldsymbol{\xi}_j \sim \mathcal{N}(\mathbf{0}, \sigma_i^2\alpha_i^2 k\rho_i\mathbf{I}) \tag{14}$$

Hence, the updated representation $\hat{z}_u$ follows a multivariate Gaussian distribution:

$$\hat{z}_u \sim \mathcal{N}\left(\sum_{i=1}^{|\mathcal{Y}|} k\rho_i\alpha_i\boldsymbol{\mu}_i, \ \sum_{i=1}^{|\mathcal{Y}|} \sigma_i^2\alpha_i^2 k\rho_i\mathbf{I}\right) \tag{15}$$

Define the total variance scalar:

$$\zeta_u := \sqrt{\sum_{i=1}^{|\mathcal{Y}|} \sigma_i^2\alpha_i^2 k\rho_i} \tag{16}$$

Using the reparameterization trick again, we can rewrite $\hat{z}_u$ as:

$$\hat{z}_u = \sum_{i=1}^{|\mathcal{Y}|} k\rho_i\alpha_i\boldsymbol{\mu}_i + \zeta_u \cdot \hat{\boldsymbol{\xi}}, \quad \text{where } \hat{\boldsymbol{\xi}} \sim \mathcal{N}(\mathbf{0}, \mathbf{I}) \tag{17}$$

Now consider the inner product with the class prototype $\boldsymbol{\mu}_c$:

$$\hat{z}_u^\top\boldsymbol{\mu}_c = \left(\sum_{i=1}^{|\mathcal{Y}|} k\rho_i\alpha_i\boldsymbol{\mu}_i\right)^\top\boldsymbol{\mu}_c + \left(\zeta_u \cdot \hat{\boldsymbol{\xi}}\right)^\top\boldsymbol{\mu}_c \tag{18}$$

By orthogonality of class means ($\boldsymbol{\mu}_i^\top\boldsymbol{\mu}_c = 1$ if $i = c$, and 0 otherwise), the first term simplifies to $k\rho_c\alpha_c$. Since $\hat{\boldsymbol{\xi}} \sim \mathcal{N}(\mathbf{0}, \mathbf{I})$, the second term becomes a linear combination of i.i.d. standard Gaussians:

$$\hat{z}_u^\top\boldsymbol{\mu}_c = k\rho_c\alpha_c + \zeta_u\sum_{i=1}^{d} \mu_{c,i}\hat{\xi}_i, \quad \hat{\xi}_i \sim \mathcal{N}(0, 1) \tag{19}$$

Because $\|\boldsymbol{\mu}_c\|_2^2 = 1$, the sum $\sum_{i=1}^{d} \mu_{c,i}\hat{\xi}_i$ is distributed as $\mathcal{N}(0, 1)$. Thus:

$$\hat{z}_u^\top\boldsymbol{\mu}_c \sim \mathcal{N}(k\rho_c\alpha_c, \zeta_u^2) \tag{20}$$

To compute the classification probability, we consider a similarity-based classifier that predicts correctly if:

$$\hat{z}_u^\top\boldsymbol{\mu}_c \geq \delta_c \tag{21}$$

Since $\hat{z}_u^\top\boldsymbol{\mu}_c$ is Gaussian distributed, we apply the cumulative distribution function of the normal distribution. For $X \sim \mathcal{N}(\mu, \sigma^2)$, we have:

$$P(X \geq a) = 1 - \Phi\left(\frac{a - \mu}{\sigma}\right) \tag{22}$$

Applying this to our case:

$$P\left(\hat{z}_u^\top\boldsymbol{\mu}_c \geq \delta_c\right) = 1 - \Phi\left(\frac{\delta_c - k\rho_c\alpha_c}{\zeta_u}\right) = 1 - \Phi\left(\frac{\delta_c - k\rho_c\alpha_c}{\sqrt{\sum_{i=1}^{|\mathcal{Y}|} \sigma_i^2\alpha_i^2 k\rho_i}}\right) \tag{23}$$

which concludes the proof. □

## A.2 Proof of Theorem A.2

*Proof.* From Theorem A.1, we know:

$$\hat{z}_u^\top\boldsymbol{\mu}_c \sim \mathcal{N}\left(k\rho_c\alpha_c, \ \sum_{i=1}^{|\mathcal{Y}|} k\rho_i\alpha_i^2\sigma_i^2\right) \tag{24}$$

and the classification probability is:

$$P\left(\hat{\boldsymbol{z}}_u^\top \boldsymbol{\mu}_c \geq \delta_c\right) = 1 - \Phi\left(\frac{\delta_c - k\rho_c\alpha_c}{\sqrt{\sum_{i=1}^{|\mathcal{Y}|} k\rho_i\alpha_i^2\sigma_i^2}}\right) \tag{25}$$

Let us denote the variance as:

$$\text{Var} := \sum_{i=1}^{|\mathcal{Y}|} k\rho_i\alpha_i^2\sigma_i^2 = k\rho_c\alpha_c^2\sigma_c^2 + \sum_{i\neq c} k\rho_i\alpha_i^2\sigma_i^2 \tag{26}$$

Let $\sigma_m^2 := \max_{i\neq c}\sigma_i^2$. Since $\alpha_i \leq \frac{1}{k}$ for $i \neq c$, we have $\alpha_i^2 \leq \frac{1}{k^2}$, and hence:

$$\sum_{i\neq c} k\rho_i\alpha_i^2\sigma_i^2 \leq \sum_{i\neq c} k\rho_i \cdot \frac{1}{k^2} \cdot \sigma_m^2 = \frac{\sigma_m^2}{k}\sum_{i\neq c}\rho_i = \frac{1-\rho_c}{k} \cdot \sigma_m^2 \tag{27}$$

Therefore:

$$\text{Var} \leq k\rho_c\alpha_c^2\sigma_c^2 + \frac{1-\rho_c}{k} \cdot \sigma_m^2 \tag{28}$$

We now use the fact that the Gaussian CDF $\Phi(z)$ is strictly increasing. Under the assumption that $\delta_c - k\rho_c\alpha_c < 0$, the numerator is negative. In this case, increasing the denominator (i.e., the variance) reduces the absolute value of $z$, making $z$ less negative. As a result, $\Phi(z)$ increases, and the classification probability $1 - \Phi(z)$ decreases.

This gives the **lower bound**:

$$P\left(\hat{\boldsymbol{z}}_u^\top \boldsymbol{\mu}_c \geq \delta_c\right) \geq 1 - \Phi\left(\frac{\delta_c - k\rho_c\alpha_c}{\sqrt{k\rho_c\alpha_c^2\sigma_c^2 + \frac{1-\rho_c}{k} \cdot \sigma_m^2}}\right) \tag{29}$$

To obtain the **upper bound**, we lower bound the total variance by dropping the non-negative cross-class terms:

$$\sum_{i\neq c} k\rho_i\alpha_i^2\sigma_i^2 \geq 0 \tag{30}$$

which leads to:

$$\text{Var} \geq k\rho_c\alpha_c^2\sigma_c^2 \tag{31}$$

Since the CDF $\Phi(z)$ is increasing and the numerator is negative, a smaller denominator results in a more negative standardized score, and hence a larger classification probability $1 - \Phi(z)$. Therefore, this yields the following **upper bound**:

$$P(\hat{\boldsymbol{z}}_u^\top \boldsymbol{\mu}_c \geq \delta_c) \leq 1 - \Phi\left(\frac{\delta_c - k\rho_c\alpha_c}{\sqrt{k\rho_c\alpha_c^2\sigma_c^2}}\right) \tag{32}$$

Combining both bounds gives the result. $\qquad\square$

### A.3  Proof of Proposition A.3

*Proof.* We analyze the monotonicity of both the upper and lower bounds in Equation 9 by taking partial derivatives of the normalized score with respect to each variable. Let the standardized score be denoted as $f(\cdot)$.

**Upper bound:**

$$f_{\text{upper}}(x) = \frac{\delta_c - k\rho_c\alpha_c}{\sqrt{k\rho_c\alpha_c^2\sigma_c^2}}$$

**(1) With respect to $k$:** Let $C_1 = \delta_c$, $C_2 = \rho_c\alpha_c$, $C_3 = \rho_c\alpha_c^2\sigma_c^2$. Then:

$$f_{\text{upper}}(k) = \frac{C_1 - kC_2}{\sqrt{kC_3}}, \quad f'_{\text{upper}}(k) = \frac{-kC_2C_3 - C_1C_3}{2(kC_3)^{3/2}} < 0 \tag{33}$$

**(2) With respect to $\rho_c$:** Let $C_2 = k\alpha_c$, $C_3 = k\alpha_c^2\sigma_c^2$. Then:

$$f_{\text{upper}}(\rho_c) = \frac{C_1 - \rho_c C_2}{\sqrt{\rho_c C_3}}, \quad f'_{\text{upper}}(\rho_c) = \frac{-\rho_c C_2 C_3 - C_1 C_3}{2(\rho_c C_3)^{3/2}} < 0 \tag{34}$$

**(3) With respect to $\alpha_c$:** Let $C_2 = k\rho_c$, $C_3 = k\rho_c\sigma_c^2$. Then:

$$f_{\text{upper}}(\alpha_c) = \frac{C_1 - C_2\alpha_c}{\alpha_c\sqrt{C_3}}, \quad f'_{\text{upper}}(\alpha_c) = \frac{-C_1}{\alpha_c^2\sqrt{C_3}} < 0 \tag{35}$$

**Lower bound:**

$$f_{\text{lower}}(x) = \frac{\delta_c - k\rho_c\alpha_c}{\sqrt{k\rho_c\alpha_c^2\sigma_c^2 + \frac{1-\rho_c}{k}\sigma_m^2}}$$

**(1) With respect to $k$:**

Let $C_1 = \delta_c$, $C_2 = \rho_c\alpha_c$, $C_3 = \rho_c\alpha_c^2\sigma_c^2$, and $C_4 = (1 - \rho_c)\sigma_m^2$. Then the lower bound becomes:

$$f_{\text{lower}}(k) = \frac{C_1 - kC_2}{\sqrt{kC_3 + \frac{C_4}{k}}} \tag{36}$$

Let $D(k) = \sqrt{kC_3 + \frac{C_4}{k}}$. Then, using the quotient and chain rule, we obtain:

$$f'_{\text{lower}}(k) = \frac{-2C_2 D^2(k) - \left(C_3 - \frac{C_4}{k^2}\right)(C_1 - kC_2)}{2D^3(k)} \tag{37}$$

Now we simplify the numerator:

$$\begin{aligned}
&- 2C_2 D^2(k) - \left(C_3 - \frac{C_4}{k^2}\right)(C_1 - kC_2) \\
=& - 2C_2 \left(kC_3 + \frac{C_4}{k}\right) - \left(C_3 - \frac{C_4}{k^2}\right)(C_1 - kC_2) \\
=& - kC_2 C_3 - \frac{3C_2 C_4}{k} - C_1 C_3 + \frac{C_1 C_4}{k} \\
=& - kC_2 C_3 - C_1 C_3 + \frac{C_4(C_1 - 3kC_2)}{k}
\end{aligned} \tag{38}$$

Under the assumption that the classifier is well-trained, we have:

$$C_1 - 3kC_2 = \delta_c - 3k\rho_c\alpha_c < \delta_c - k\rho_c\alpha_c < 0$$

Hence, each term in Equation (38) is negative, and the overall numerator is negative. Since the denominator $2D^3(k) > 0$, we conclude:

$$f'_{\text{lower}}(k) < 0 \tag{39}$$

which means the normalized score decreases with $k$, and thus the classification probability increases.

**(2) With respect to $\rho_c$:**

Let $C_1 = \delta_c$, $C_2 = k\alpha_c$, $C_3 = k\alpha_c^2\sigma_c^2$, and $C_4 = \sigma_m^2$. Then the lower bound becomes:

$$f_{\text{lower}}(\rho_c) = \frac{C_1 - C_2\rho_c}{\sqrt{C_3\rho_c + \frac{C_4(1-\rho_c)}{k}}} \tag{40}$$

Let $D(\rho_c) := \sqrt{C_3\rho_c + \frac{C_4(1-\rho_c)}{k}}$. Then using the chain rule, the derivative is:

$$f'_{\text{lower}}(\rho_c) = \frac{-2C_2 D^2(\rho_c) - \left(C_3 - \frac{C_4}{k}\right)(C_1 - C_2\rho_c)}{2D^3(\rho_c)} \tag{41}$$

We now simplify the numerator:

$$-2C_2D^2(\rho_c) - \left(C_3 - \frac{C_4}{k}\right)(C_1 - C_2\rho_c)$$

$$= -2C_2\left(C_3\rho_c + \frac{C_4(1-\rho_c)}{k}\right) - \left(C_3 - \frac{C_4}{k}\right)(C_1 - C_2\rho_c)$$

$$= -2C_2C_3\rho_c - \frac{2C_2C_4(1-\rho_c)}{k} - C_3C_1 + \frac{C_4C_1}{k} + C_2\rho_c C_3 - \frac{C_2\rho_c C_4}{k}$$

$$= -C_2C_3\rho_c - C_1C_3 - \frac{2C_2C_4}{k} + \frac{C_2\rho_c C_4}{k} + \frac{C_1C_4}{k} \tag{42}$$

Hence the full numerator is:

$$-C_2C_3\rho_c - C_1C_3 + \frac{C_4(C_2\rho_c + C_1 - 2C_2)}{k} \tag{43}$$

We now verify its sign. Under the assumption that the classifier is well-trained, i.e.,

$$C_1 - C_2\rho_c = \delta_c - k\rho_c\alpha_c < 0 \quad \Rightarrow \quad C_1 < C_2\rho_c$$

which implies:

$$C_1 + C_2\rho_c < 2C_2\rho_c < 2C_2 \quad \Rightarrow \quad C_1 + C_2\rho_c - 2C_2 < 0$$

Thus the entire numerator is negative, and since $D(\rho_c) > 0$, we conclude:

$$f'_{\text{lower}}(\rho_c) < 0 \tag{44}$$

That is, the normalized score decreases as $\rho_c$ increases, and hence the classification probability increases.

### (3) With respect to $\alpha_c$:

Let $C_1 = \delta_c$, $C_2 = k\rho_c$, $C_3 = k\rho_c\sigma_c^2$, and $C_4 = \frac{1-\rho_c}{k}\sigma_m^2$. Then the lower bound becomes:

$$f_{\text{lower}}(\alpha_c) = \frac{C_1 - C_2\alpha_c}{\sqrt{C_3\alpha_c^2 + C_4}} \tag{45}$$

Let $D(\alpha_c) := \sqrt{C_3\alpha_c^2 + C_4}$. Applying the quotient and chain rule, we obtain:

$$f'_{\text{lower}}(\alpha_c) = \frac{-C_2D^2(\alpha_c) - C_3\alpha_c(C_1 - C_2\alpha_c)}{D^3(\alpha_c)} \tag{46}$$

We now expand the numerator:

$$-2C_2(C_3\alpha_c^2 + C_4) - C_3\alpha_c(C_1 - C_2\alpha_c)$$

$$= -2C_2C_3\alpha_c^2 - 2C_2C_4 - C_1C_3\alpha_c + C_2C_3\alpha_c^2$$

$$= -C_2C_3\alpha_c^2 - 2C_2C_4 - C_1C_3\alpha_c \tag{47}$$

All three terms in the numerator are negative, and the denominator is strictly positive. Therefore:

$$f'_{\text{lower}}(\alpha_c) < 0 \tag{48}$$

This implies that the normalized score decreases as $\alpha_c$ increases, and hence the classification probability increases.

### (4) With respect to variances $\sigma_c$ and $\sigma_m$:

In both bounds, $\sigma_c$ and $\sigma_m$ appear only in the denominator. Increasing either of them increases the total variance, which increases the normalized score $f(x)$ (i.e., makes it less negative), and hence decreases the classification probability $1 - \Phi(f(x))$.

**Conclusion:** In both upper and lower bounds, the classification probability is monotonically increasing with respect to $k$, $\rho_c$, and $\alpha_c$, and monotonically decreasing with respect to $\{\sigma_i\}_{i=1}^{|\mathcal{Y}|}$. $\qquad\square$

## A.4 Proof of Proposition 4.1

*Proof.* Let $u, v \in \mathcal{V}$ belong to the same cluster, i.e., $\mathcal{P}(u) = \mathcal{P}(v) = p$. In the dense attention setting defined by mask $\mathbf{M}^{c3}$, the attention weight from $u$ to $v$ is given by:

$$\alpha_{uv}^{(c3)} = \frac{\exp(\langle \mathbf{q}_u, \mathbf{k}_v \rangle)}{\sum_{v' \in \mathcal{P}^{-1}(p)} \exp(\langle \mathbf{q}_u, \mathbf{k}_{v'} \rangle)}$$

Now consider the two-layer attention structure using mask $\mathbf{M}^{c4}$. In the second layer, node $u$ attends to:

- Itself, with attention score $\alpha_{uu}^{(2)}$

- Its virtual cluster node $p$, with attention score $\alpha_{up}^{(2)}$

The virtual node $p$ in the first layer aggregates from all nodes in cluster $p$, including $v$, with attention score $\alpha_{pv}^{(1)}$.

Therefore, the total contribution of node $v$ to node $u$ through the two-layer structure can be approximated as:

$$\alpha_{uv}^{(c3)} \approx \begin{cases} \alpha_{uu}^{(2)} + \alpha_{up}^{(2)} \cdot \alpha_{pv}^{(1)}, & \text{if } u = v \\ \alpha_{up}^{(2)} \cdot \alpha_{pv}^{(1)}, & \text{if } u \neq v \end{cases}$$

This formulation shows that the dense cluster-wise attention score can be decomposed into a mixture of self-attention and two-hop attention via the cluster-level virtual node. This completes the proof. $\square$

## A.5 Proof of Propostion 4.2

*Proof.* As detailed in Appendix B, the space complexity of the dense attention computation is $O(2hN^2)$, while that of the sparse computation is $O(6hmd_h)$. By comparing the two, we conclude that the sparse scheme is more memory-efficient when:

$$\frac{m}{N^2} < \frac{1}{3d_h} \quad \Leftrightarrow \quad \kappa_{\mathcal{R}_i} < \frac{1}{3d_h}$$

where $\kappa_{\mathcal{R}_i} := \frac{m}{N^2}$ denotes the relative sparsity of the receptive field. This completes the proof. $\square$

# B Efficiency Comparison of Attention Computation Schemes

We provide the pseudocode for masked multi-head attention with dense computation scheme in Algorithm 1. As illustrated, the primary memory overhead stems from storing the raw attention scores $\mathbf{S}$ and the normalized attention weights $\mathbf{A}$, both of which have a space complexity of $O(hN^2)$. In total, the algorithm requires $O(2hN^2)$ memory. In terms of time complexity, the main cost comes from three parts: (1) the linear transformations for generating the QKV matrices, which require $O(3Nd^2)$; (2) the computation of the attention matrix, $O(N^2d)$; and (3) the multiplication between the attention matrix and the value matrix, $O(N^2d)$. Summing up, the total time complexity of standard MHA is $O(3Nd^2 + 2N^2d)$.

---

**Algorithm 1** Masked Multi-Head Attention

---

**Require:** Input $\mathbf{Z} \in \mathbb{R}^{N \times d_{\text{model}}}$, number of heads $h$, projection matrices $\mathbf{W}_Q$, $\mathbf{W}_K$, $\mathbf{W}_V$, binary attention mask $\mathbf{M} \in \{0, 1\}^{N \times N}$
**Ensure:** Output $\mathbf{Y} \in \mathbb{R}^{N \times d_{\text{model}}}$
 1: Project input to queries, keys, values:

$$\mathbf{Q}, \mathbf{K}, \mathbf{V} = \mathbf{Z}\mathbf{W}_Q, \ \mathbf{Z}\mathbf{W}_K, \ \mathbf{Z}\mathbf{W}_V \in \mathbb{R}^{N \times d_{\text{model}}}$$

 2: Reshape and split into $h$ heads:

$$\mathbf{Q}, \mathbf{K}, \mathbf{V} \in \mathbb{R}^{h \times N \times d_h} \quad \text{where } d_h = d_{\text{model}}/h$$

 3: Compute raw attention scores (per head):

$$\mathbf{S} = \frac{\mathbf{Q}\mathbf{K}^\top}{\sqrt{d_h}} \in \mathbb{R}^{h \times N \times N}$$

 4: Apply attention mask:

$$\mathbf{S}_{i,j} = \begin{cases} \mathbf{S}_{i,j}, & \text{if } \mathbf{M}_{i,j} = 1 \\ -\infty \text{ (or a large negative constant)}, & \text{if } \mathbf{M}_{i,j} = 0 \end{cases}$$

 5: Normalize via Softmax:

$$\mathbf{A} = \text{softmax}(\mathbf{S}) \in \mathbb{R}^{h \times N \times N}$$

 6: Apply attention weights:

$$\mathbf{H} = \mathbf{A}\mathbf{V} \in \mathbb{R}^{h \times N \times d_h}$$

 7: Concatenate heads as the final output:

$$\mathbf{Y} = \text{Concat}_{\text{head}}(\mathbf{H}) \in \mathbb{R}^{N \times d_{\text{model}}}$$

 8: **return Y**

---

The sparse attention computation for head $i$ can be formulated as:

$$\text{head}_i(\mathbf{H}, \mathbf{M})_u = \sum_{v \in \mathcal{M}_u} \frac{\exp\left((\mathbf{H}_u \mathbf{W}_Q^{(i)})(\mathbf{H}_v \mathbf{W}_K^{(i)})^\top\right)}{\sum_{t \in \mathcal{M}_u} \exp\left((\mathbf{H}_u \mathbf{W}_Q^{(i)})(\mathbf{H}_t \mathbf{W}_K^{(i)})^\top\right)} \mathbf{H}_v \mathbf{W}_V^{(i)} \tag{49}$$

where $\mathcal{M}_u = \{v : \mathbf{M}_{u,v} = 1\}$ denote the key set of node $u$, $\mathbf{H}_u, \mathbf{H}_v \in \mathbb{R}^{1 \times d}$ denote the representations of node $u, v$, and $\mathbf{W}_Q^{(i)}, \mathbf{W}_K^{(i)}, \mathbf{W}_V^{(i)} \in \mathbb{R}^{d \times d_h}$ are the projection matrices for the $i$-th head. The outputs of all heads are concatenated to produce the final output: $\text{MHA}^S(\mathbf{H}, \mathbf{M}) = \text{Concat}(\text{head}_1, \ldots, \text{head}_H)$.

Next, we present the pseudocode for Sparse Multi-Head Attention in Algorithm 2. The dominant memory consumption arises from storing intermediate variables $\mathbf{Q}'$, $\mathbf{K}'$, $\mathbf{S}'$, $\mathbf{V}'$, and $\mathbf{H}'$, each contributing to a space complexity of $O(mhd_h)$, resulting in a total of $O(5mhd_h)$. Additionally, computing the output $\mathbf{H}$ involves a $\text{Scatter}_{\text{sum}}(\cdot)$ operation, which requires an auxiliary buffer of size $O(mhd_h)$. Therefore, the overall space complexity of Sparse Multi-Head Attention is $O(6mhd_h)$. Regarding time complexity, sparse MHA consists of three main parts: (1) feature transformations, $O(3Nd^2)$; (2) sparse attention score computation (Step 4), $O(2md)$; and (3) output computation (Step 6), $O(2md)$. Therefore, the total time complexity is $O(3Nd^2 + 4md)$. A direct comparison indicates that sparse MHA outperforms standard MHA when $\frac{m}{N^2} < \frac{1}{2}$.

According to our "Dual Attention Computation Scheme" in Section 4.4 and Proposition 4.2, we apply sparse attention computation in most regions, except for attention from global nodes to the origin node, which involves a dense attention region (where $\frac{m}{N^2} = 1$, as shown in Figure 2). This hybrid design allows our method to achieve lower overall time complexity than both standard and sparse MHA, i.e., $\min(O(3Nd^2 + 2N^2d), O(3Nd^2 + 4md))$.

**Algorithm 2** Sparse Multi-Head Attention

---

**Require:** Input representation matrix $\mathbf{Z} \in \mathbb{R}^{N \times d_{\text{model}}}$, sparse index mask $\mathbf{M} \in \mathbb{Z}_+^{2 \times m}$, where $m$ is the number of non-zero entries; projection matrices $\mathbf{W}_Q, \mathbf{W}_K, \mathbf{W}_V \in \mathbb{R}^{d_{\text{model}} \times d_{\text{model}}}$; number of heads $h$

**Ensure:** Output $\mathbf{Y} \in \mathbb{R}^{N \times d_{\text{model}}}$

1: Linear projections:

$$\mathbf{Q} = \mathbf{Z}\mathbf{W}_Q, \quad \mathbf{K} = \mathbf{Z}\mathbf{W}_K, \quad \mathbf{V} = \mathbf{Z}\mathbf{W}_V \quad \in \mathbb{R}^{N \times d_{\text{model}}}$$

2: Reshape for multi-head:

$$\mathbf{Q}, \mathbf{K}, \mathbf{V} \in \mathbb{R}^{N \times h \times d_h}, \quad d_h = d_{\text{model}}/h$$

3: Index via sparse mask:

$$\mathbf{Q}' = \mathbf{Q}[\mathbf{M}_0], \quad \mathbf{K}' = \mathbf{K}[\mathbf{M}_1] \quad \in \mathbb{R}^{m \times h \times d_h}$$

4: Sparse attention score computation:

$$\mathbf{S}' = \mathbf{Q}' \cdot \mathbf{K}' \in \mathbb{R}^{m \times h \times d_h}$$

$$\mathbf{S} = \text{SUM}(\mathbf{S}', \text{dim} = -1) \in \mathbb{R}^{m \times h}$$

5: Apply softmax normalization:

$$\tilde{\mathbf{S}} = \exp(\mathbf{S}) \in \mathbb{R}^{m \times h}$$

$$\tilde{\mathbf{S}}_{\text{sum}} = \text{Scatter}_{\text{sum}}(\tilde{\mathbf{S}}, \ \mathbf{M}_0, \ \text{dim} = 0) \in \mathbb{R}^{N \times h}$$

$$\mathbf{A} = \tilde{\mathbf{S}}/\tilde{\mathbf{S}}_{\text{sum}}[\mathbf{M}_0] \in \mathbb{R}^{m \times h}$$

6: Aggregate weighted values:

$$\mathbf{V}' = \mathbf{V}[\mathbf{M}_1] \in \mathbb{R}^{m \times h \times d_h}$$

$$\mathbf{H}' = \mathbf{A} \cdot \mathbf{V}' \in \mathbb{R}^{m \times h \times d_h}$$

$$\mathbf{H} = \text{Scatter}_{\text{sum}}(\mathbf{H}', \ \mathbf{M}_0, \ \text{dim} = 0) \in \mathbb{R}^{N \times h \times d_h}$$

7: Concatenate heads as the final output:

$$\mathbf{Y} = \text{Concat}_{\text{head}}(\mathbf{H}) \in \mathbb{R}^{N \times d_{\text{model}}}$$

8: **return** $\mathbf{Y}$

---

## C  The Summary Table of GTs and Hierarchical Attention Masks

We summarize many Graph Transformers and their corresponding hierarchical attention masks mentioned in Section 3 in Table 4.

Table 4: Summary of Graph Transformers and their corresponding hierarchical attention mask.

| Mask Type | Mask Notation | Representative Graph Transformers |
|---|---|---|
| Local Masks | $\mathbf{M}^{l1}$ | GOAT[24], NAGphormer[5], VCR-Graphormer[15], GCFormer[6] |
| | $\mathbf{M}^{l2}$ | NodeFormer[42], SGFormer[43], PolyNormer[11], CoBFormer[44], NAGphormer[5], VCR-Graphormer[15], and GCFormer[6] |
| Cluster Masks | $\mathbf{M}^{c1}$ | Graph ViT[18], Cluster-GT[20], CoBFormer[44] |
| | $\mathbf{M}^{c2}$ | Cluster-GT[20] |
| | $\mathbf{M}^{c3}$ | CoBFormer[44] |
| Global Masks | $\mathbf{M}^{g1}$ | NodeFormer[42], SGFormer[43], PolyNormer[11] |
| | $\mathbf{M}^{g2}$ | Exphormer[34] |

# D Dataset

Table 5: The detailed dataset statistics.

| Dataset | #Nodes | #Edges | #Feats | Edge hom | #Classes |
|---|---|---|---|---|---|
| Cora | 2,708 | 5,429 | 1,433 | 0.83 | 7 |
| CiteSeer | 3,327 | 4,732 | 3,703 | 0.72 | 6 |
| PubMed | 19,717 | 44,338 | 500 | 0.79 | 3 |
| Photo | 7,650 | 119,081 | 745 | 0.83 | 8 |
| Computer | 13,752 | 245,861 | 767 | 0.78 | 10 |
| Squirrel | 2,223 | 46,998 | 2,089 | 0.21 | 5 |
| Chameleon | 890 | 8,854 | 2,325 | 0.24 | 5 |
| Minesweeper | 10,000 | 39,402 | 7 | 0.68 | 2 |
| Ogbn-Arxiv | 169,343 | 1,166,343 | 128 | 0.63 | 40 |

## D.1 Dataset Statistics

The detailed dataset statistics are listed in Table 5. The edge homophily is defined as:

$$h = \frac{|u, v : \mathbf{y}_u = \mathbf{y}_v|}{E}$$

The Cora, CiteSeer, PubMed [45], Photo, and Computer [33] datasets are available through PyG [14], while Ogbn-Arxiv can be accessed via the OGB platform [19]. The Chameleon, Squirrel, and Minesweeper datasets are provided in the official repository of [31].

## D.2 Dataset Splitting Protocol

For Computer and Photo, we follow the splitting protocol in [11, 15], randomly dividing nodes into training, validation, and test sets with a 60%:20%:20% ratio over five runs. For Ogbn-Arxiv, we adopt the official split provided in [19]. The remaining datasets are split into 50%:25%:25% train/validation/test sets, repeated five times following [42, 31].

# E Baselines

We compare M$^3$Dphormer against 15 baselines spanning multiple model families:

1) *Classic GNNs*:

- **GCN** [23] adopts a spectral-based convolution that aggregates and transforms features from immediate neighbors using a normalized adjacency matrix. It can be interpreted as a form of Laplacian smoothing.
- **GAT** [38] introduces a self-attention mechanism to assign learnable weights to different neighbors, enabling adaptive and context-aware feature aggregation.
- **SAGE** [16] is an inductive framework that samples a fixed-size neighborhood and aggregates features through functions such as mean, LSTM, or pooling, allowing generalization to unseen nodes and efficient training on large graphs.

2) *Enhanced Classic GNNs*: **GCN\***, **GAT\***, and **SAGE\*** are strong baselines proposed in [26], a benchmark study showing that classic GNNs can achieve significantly better performance on node classification tasks by careful hyperparameter tuning and the incorporation of advanced training techniques, such as residual connections [17] and normalization methods [48, 2, 21].

3) *Advanced GNNs*: To move beyond the widely adopted homophily assumption—that nodes of the same class are more likely to be connected [28]—advanced GNNs such as GPRGNN [7] and FAGCN [3] have been proposed to improve performance on heterophilic graphs.

- **GPRGNN** [7] employs a generalized PageRank (GPR) propagation scheme, which allows flexible and learnable weighting over multi-hop neighborhood information. This design enables the model to adapt to both homophilic and heterophilic graph structures.

- **FAGCN** [3] introduces a frequency adaptive mechanism that modulates the importance of low- and high-frequency components in the spectral domain. By learning a task-specific filter, FAGCN effectively balances local smoothness and discriminative power, making it suitable for graphs with varying levels of heterophily.

4) *SOTA Graph Transformers*:

- **NAGphormer** [5] tokenizes multi-hop neighborhoods into fixed-length sequences using a Hop2Token module, enabling scalable and efficient node classification on large graphs.
- **Exphormer** [34] designs a sparse Transformer using expander graphs and virtual global nodes, achieving linear complexity and strong performance on large-scale graphs.
- **SGFormer** [43] simplifies the Transformer architecture by adopting a shallow attentive propagation without positional encodings, ensuring efficient all-pair interactions.
- **CoBFormer** [44] mitigates over-globalization by combining coarse-grained and fine-grained paths, improving the model's balance between global and local information.
- **PolyNormer** [11] captures complex structures using polynomial-expressive attention with linear time complexity, and performs well on both homophilic and heterophilic graphs.

5) *MoE-based GNNs*:

- **Mowst** [47] introduces a Mixture of Experts (MoE) framework that combines a weak expert (MLP) and a strong expert (GNN). A confidence-based gating mechanism determines whether to activate the strong expert for each node, enabling adaptive computation and improved performance across diverse graph structures.
- **GCN-MoE** [40] applies the MoE paradigm to GCNs by incorporating multiple experts with varying neighborhood aggregation ranges. A gating function dynamically selects the appropriate expert for each node, enhancing the model's capacity to handle graphs with diverse structural patterns.

## F  Experimental Details

This section provides the detailed experimental setup corresponding to the results reported in the main paper.

### F.1  Training Strategy

We follow the training protocol used in the official implementation of NAGphormer and train it using a mini-batch strategy on all datasets. For all other baselines and our proposed M$^3$Dphormer, we adopt a full-batch training scheme. The Adam Optimizer is used for optimization.

### F.2  M$^3$Dphormer Configuration

We implement M$^3$Dphormer by stacking multiple M$^3$Dphormer layers. All hyperparameters are selected via grid search over the following search space:

- Learning rate: $\{5 \times 10^{-4}, 10^{-3}, 5 \times 10^{-3}\}$
- Number of M$^3$Dphormer layers:
  - Cora, Citeseer, Pubmed, Chameleon, Photo: $\{2, 3, 4\}$
  - Squirrel, Computer, Ogbn-Arxiv: $\{5, 6, 7\}$
  - Minesweeper: $\{10, 12, 15\}$
- Number of attention heads: $\{1, 2, 4, 8\}$
- Hidden dimension: $\{64, 128, 256\}$
- Weight decay: $\{0, 10^{-4}, 5 \times 10^{-4}, 10^{-3}, 5 \times 10^{-3}\}$
- Dropout rate: $\{0.3, 0.5, 0.7\}$
- Attention dropout rate: $\{0.1, 0.3, 0.5\}$

- Number of clusters:
    - Cora, Citeseer, Squirrel, Minesweeper: {96, 128, 160, 192}
    - Pubmed, Photo, Computer: {160, 192, 224, 256}
    - Ogbn-Arxiv: {2048}
    - Chameleon: {32, 64, 96, 128}

We apply Pre-RMSNorm [48] before the bi-level expert routing mechanism in each M$^3$Dphormer layer for most datasets. For Ogbn-Arxiv, however, we adopt Post-BatchNorm due to its superior convergence behavior. We adopt GAT-style attention for the local expert due to its computational efficiency, and standard multi-head attention (MHA) for the cluster and global experts. This choice is motivated by the observation in [4] that GAT-style attention suffers from a static attention problem, which can significantly degrade the performance of cluster and global experts.

## F.3 Baselines

We implement GCN, SAGE, GAT, GPRGNN, and FAGCN using PyG [14]. For all other baselines, we use the official implementations. The corresponding repositories are listed below:

- GCN*, GAT*, SAGE*: `https://github.com/LUOyk1999/tunedGNN`
- NAGphormer: `https://github.com/JHL-HUST/NAGphormer`
- Exphormer: `https://github.com/hamed1375/Exphormer`
- SGFormer: `https://github.com/qitianwu/SGFormer`
- CoBFormer: `https://github.com/null-xyj/CoBFormer`
- PolyNormer: `https://github.com/cornell-zhang/Polynormer`
- Mowst: `https://github.com/facebookresearch/mowst-gnn`
- GCN-MOE: `https://github.com/VITA-Group/Graph-Mixture-of-Experts`

We follow the official training protocols and perform hyperparameter tuning for each model on every dataset. The search space is defined as follows:

- Learning rate: $\{5 \times 10^{-4}, 10^{-3}, 5 \times 10^{-3}\}$
- Hidden dimension: $\{64, 128, 256\}$
- Dropout rate: $\{0.3, 0.5, 0.7\}$
- Weight decay: $\{0, 10^{-4}, 5 \times 10^{-4}, 10^{-3}, 5 \times 10^{-3}\}$

For models with additional key hyperparameters, we further tune them as follows:

- Attention-based models: number of heads $\in \{1, 2, 4, 8\}$
- NAGphormer: number of hops $\in \{3, 5, 7, 10, 15\}$
- SGFormer: $\alpha \in \{0.5, 0.8\}$
- CoBFormer: $\alpha \in \{0.9, 0.8, 0.7\}; \quad \tau \in \{0.9, 0.7, 0.5, 0.3\}$
- GCN-MOE: number of experts $\in \{3, 4, 5\}$
- FAGCN: $\epsilon \in \{0.3, 0.5, 0.7\}$
- GPRGNN: $\alpha \in \{0.1, 0.3, 0.5\}$

All models are trained on a single NVIDIA GPU with 24GB memory. We run each method 5 times and report the mean and standard deviation of the results.

Table 6: Performance and runtime comparison across datasets.

| | Cora | Citeseer | Pubmed | Photo | Computer | Chameleon | Squirrel |
|---|---|---|---|---|---|---|---|
| Dual | 5.62s | 4.92s | 6.41s | 5.34s | 10.86s | 5.58s | 12.29s |
| Dense | 6.69s | 7.28s | OOM | 7.25s | OOM | 6.28s | 16.45s |
| Sparse | 6.28s | 5.67s | 7.16s | 6.01s | 15.56s | 7.60s | 14.38s |
| Polynormer | 3.59s | 2.76s | 3.60s | 6.02s | 4.16s | 3.88s | 5.53s |

# G   More Experimental Results

## G.1   Runtime Comparison

We report the training time of our method over 200 epochs, and compare it against its sparse and dense variants as well as PolyNormer [11]. The results are summarized in Table 6. We observe that the proposed "Dual Attention Computation Scheme" achieves faster training than both sparse and dense MHA variants. Although it is marginally slower than PolyNormer, the additional cost arises from employing three MHA experts, which are crucial for attaining higher accuracy. Furthermore, our model converges within 200 epochs on most datasets, rendering the time overhead acceptable.

## G.2   Graph Classification Performance

We extend our method to graph-level tasks by incorporating edge features and Laplacian positional encodings. To assess the effectiveness of this extension, we conduct experiments on two graph classification datasets: OGBG-MOLBACE and OGBG-MOLBBBP. We compare our approach with three widely-used GNN baselines—GCN, GAT, and GINE—which are recognized as strong performers on graph-level benchmarks [27]. The results are summarized in Table 7.

Table 7: Graph classification performance($\% \pm \sigma$), measured by ROC-AUC.

| Method | ogbg-bace | ogbg-bbbp |
|---|---|---|
| $M^3$Dphormer | 0.80432±0.01040 | 0.68232±0.00632 |
| GCN* | 0.75680±0.01676 | 0.65146±0.01184 |
| GAT* | 0.78149±0.01900 | 0.65175±0.01221 |
| GINE* | 0.74799±0.01014 | 0.65095±0.01143 |

As shown in the Table 7, our extended model outperforms the baselines on both datasets, demonstrating its strong potential for graph-level tasks.

## G.3   Ablation Study on FFN Variants

In our method, the standard Transformer Feed-Forward Network (FFN) module is replaced with a single linear layer followed by an activation function, such as ReLU [1]. To investigate the impact of the FFN module on node classification performance, we compare $M^3$Dphormer with its FFN variants. The results are summarized in Table 8.

Table 8: Node classification results of M3Dphormer and its FFN variant.

| | Cora | Citeseer | Pubmed | Photo | Computer | Chameleon | Squirrel |
|---|---|---|---|---|---|---|---|
| M3Dphormer | 88.48±1.94 | 77.53±1.56 | 89.96±0.49 | 95.91±0.68 | 92.09±0.46 | 47.09±4.05 | 44.34±1.94 |
| +FFN | 86.50±1.87 | 75.53±2.19 | 89.28±0.30 | 95.37±0.54 | 91.28±0.72 | 44.22±2.81 | 41.36±1.82 |

As observed, incorporating the FFN module often results in a noticeable performance drop, particularly on datasets such as Cora, Citeseer, Chameleon, and Squirrel. We attribute this to the relatively simple and easily learnable node features in these datasets. In such cases, capturing complex node interactions becomes more critical, a task that is more effectively handled by the Multi-Head Attention

(MHA) module. While MHA is commonly regarded as the core of Transformer architectures, it is important to note that the standard FFN module typically contains twice as many parameters as MHA (e.g., with a projection of $d \to 4d \to d$). Since meaningful structural interactions are primarily modeled through attention, it is natural to allocate greater capacity to the MHA module.

In addition to the observed performance improvement, reducing or simplifying the FFN module can significantly enhance computational and memory efficiency, making the model more lightweight and scalable.

### G.4   Loss and Accuracy Curves For PolyNormer

We plot the accuracy and loss curves of M³Dphormer and PolyNormer[11] in Figure 6. Across the training, validation, and test sets, M³Dphormer consistently achieves faster convergence and higher accuracy than PolyNormer during the training process, demonstrating the effectiveness of our method.

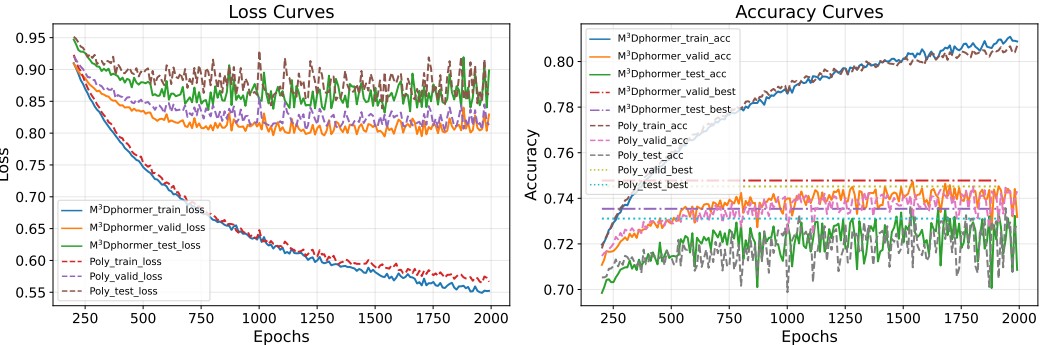

Figure 6: Comparison of accuracy and loss curves between M³DFormer and PolyNormer on Arxiv.

### G.5   More Visualization Results

We visualize the distribution of learned gate weights across node degree bins on Ogbn-Arxiv in Figure 7. Several observations can be made: 1) In the first layer, the gate weights remain close to the initialization $[0.5, \ 0.25, \ 0.25]$, suggesting that all types of interactions are considered at the early stage. 2) In the second layer, the weights for local and cluster experts increase, indicating a shift in focus toward capturing structural information at local and mid-range levels. 3) In the third and fourth layers, local experts consistently dominate across all degree bins. Meanwhile, the gate weights for cluster and global experts exhibit a decreasing trend with increasing node degree, reflecting the tendency of low-degree nodes—often located near cluster boundaries with low local homophily—to rely more on broader contextual information. 4) In the final layer, we observe a notable shift: the weights assigned to local experts decrease as node degree increases, while those for global experts increase. This trend may be attributed to the fact that high-degree nodes have already aggregated sufficient local information, and further local aggregation may lead to over-smoothing [25, 29, 30] or over-squashing [36, 10].

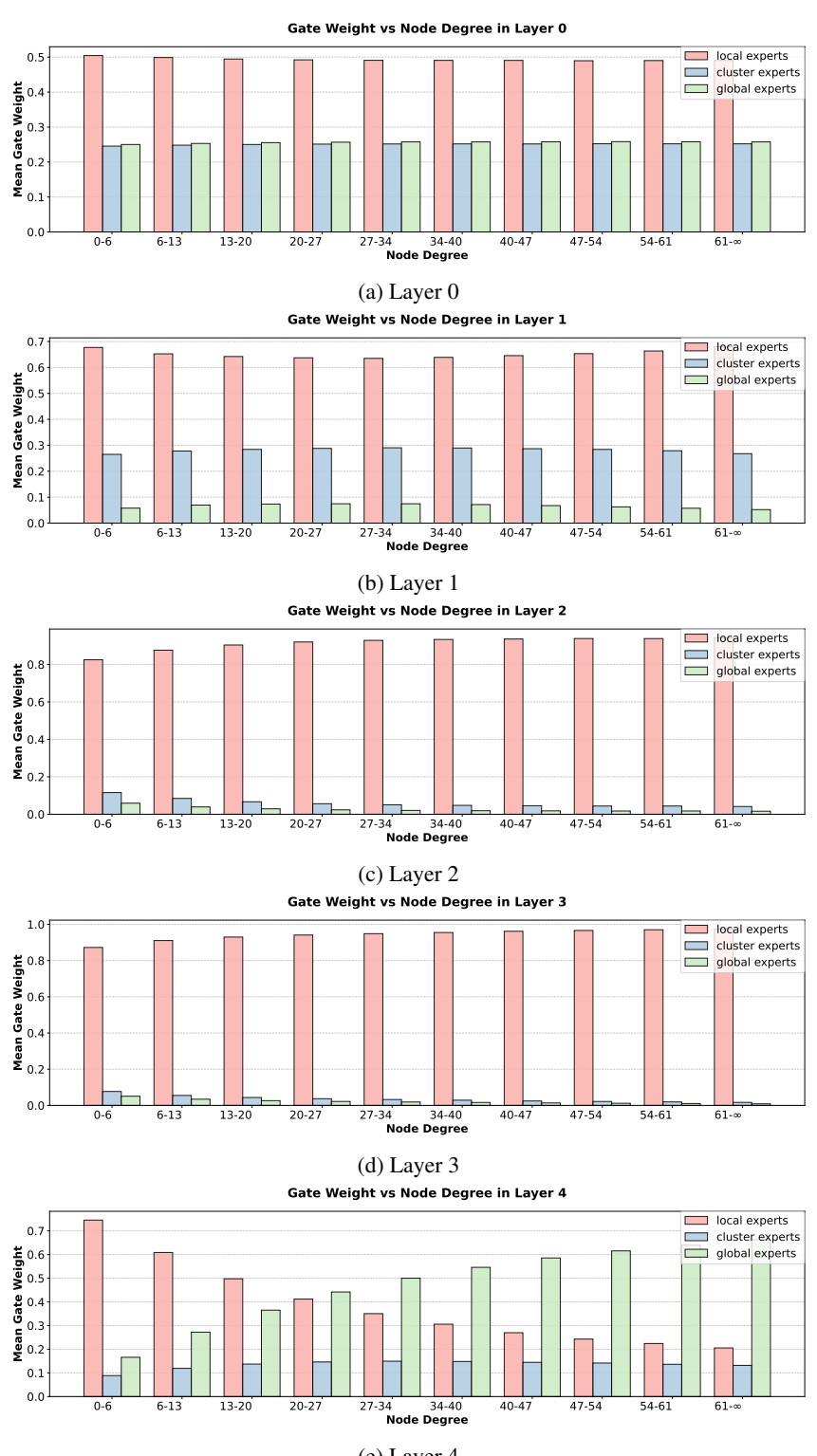

(a) Layer 0

(b) Layer 1

(c) Layer 2

(d) Layer 3

(e) Layer 4

Figure 7: Gate weight distribution across node degree bins for different layers.

