# OpenReview forum: "Unifying and Enhancing Graph Transformers via a Hierarchical Mask Framework"
_NeurIPS.cc/2025/Conference — NeurIPS 2025 poster_

### Official Review · Reviewer_rvAf · 2025-06-29

**Clarity:** 3
**Significance:** 4
**Originality:** 4
**Rating:** 5
**Confidence:** 5

**Summary:**

This paper revisits Graph Transformers through the lens of hierarchical interactions and proposes a unified hierarchical mask framework, which models different interaction levels via attention masks rather than architectural modifications. This design simplifies the model space and promotes structural consistency. Theoretical analysis under this framework identifies two key factors affecting node classification — receptive field size and label consistency — and argues that no single interaction level suffices across all scenarios. This motivates the integration of multiple levels of interaction. To address this, the authors propose M$^3$DPhormer, which uses three hierarchical masks, a bi-level expert routing mechanism for adaptive fusion, and a dual attention computation scheme for scalability. Experimental results across benchmarks demonstrate the effectiveness of both the framework and the proposed model.

**Questions:**

1.	Given that the main difference between experts lies in their attention masks, have the authors considered sharing parameters across the MHA modules to improve parameter efficiency?
2.	The authors state that node classification performance is positively correlated with label consistency. Could the authors clarify how label consistency differs from the homophily rate?

**Ethical Concerns:**

["NO or VERY MINOR ethics concerns only"]

**Final Justification:**

The authors' rebuttal has address my concern. Based on the comments of all reviewers, I'd like to keep my score.

**Limitations:**

YES

**Quality:**

4

**Strengths And Weaknesses:**

Strengths:
1. The paper is well-organized. It starts with a solid review of existing Graph Transformers and introduces a unified hierarchical mask framework. Both the theory and experiments are well-aligned, clearly identifying core challenges in modeling hierarchical interactions and proposing targeted solutions through M$^3$DPhormer.
2. The framework is valuable in that it shifts focus away from complex architectures, showing that diverse interactions can be captured more cleanly via attention mask design. This abstraction also makes the theoretical analysis more tractable and informative for future model design.
 3. M$^3$DPhormer is a thoughtful and well-motivated model. The bi-level expert routing supports adaptive multi-level fusion, while the dual attention scheme improves scalability. Experimental results are strong and provide convincing support for the proposed method.

Weaknesses:
1. More details about the proposed method should be clarified. In particular, it would be helpful to explain how the number of clusters is determined when constructing the cluster-level mask, and how the appropriate computation scheme is selected for different regions within the masks.

---

> ### Author Rebuttal · Authors · 2025-07-30
>
> We are immensely grateful for the reviewer's constructive and valuable feedback. We are dedicated to thoroughly examining each issue raised and providing detailed responses to enhance our paper.
>
> **More Details**
>
> We have provided detailed experimental settings in Appendix G, and an anonymous ZIP archive containing our code is included in the Supplementary Material. You may refer to Appendix G for specific experimental configurations, and reproduce our results by running the `runs.sh` script included in the provided code.
>
> Regarding the selection of the number of clusters, we preset a range based on the total number of nodes in each dataset to ensure that the cluster sizes remain appropriate. The selected cluster numbers are then treated as a tunable hyperparameter. The specific candidate cluster numbers for each dataset are listed in Appendix G.2. As shown in the parameter analysis in Figure 4 and Appendix H.2, the model’s performance is generally robust with respect to the number of clusters in most scenarios.
>
> For the design of computation schemes in different regions, we follow the theoretical guidance provided by Proposition 4.2. Specifically, we use sparse attention for the local (local masks), intra-cluster (cluster masks), and node-to-global (global mask) regions, where the low density of non-zero entries makes sparse computation more efficient. In contrast, we use dense attention for the global-to-node region (also in the global mask), as this region is fully connected (all ones), making dense computation more efficient in practice. Experimental results in Figure 3 and Appendix H.1 empirically confirm the efficiency benefits of this design.
>
> **Sharing parameters across experts**
>
> We conducted an experiment where the parameters of the three attention experts were shared. The results are presented in the table below:
>
> |                  | cora       | citeseer   | pubmed     | photo      | computer   | chameleon  | squirrel   |
> | ---------------- | ---------- | ---------- | ---------- | ---------- | ---------- | ---------- | ---------- |
> | our              | 88.48±1.94 | 77.53±1.56 | 89.96±0.49 | 95.91±0.68 | 92.09±0.46 | 47.09±4.05 | 44.34±1.94 |
> | share parameters | 87.06±1.93 | 76.69±1.62 | 89.14±0.48 | 95.24±0.57 | 91.77±0.37 | 43.68±3.79 | 40.50±1.51 |
>
> These results show that sharing parameters among the three experts leads to a noticeable drop in performance across most datasets. We believe this is because each expert is designed to capture interaction information at a different level. When the same parameters are shared among them, each expert is forced to handle multi-level information simultaneously, which may cause conflicts in the learning process and hinder effective representation learning.
>
> **Homophily rate and label consistency**
>
> While the homophily rate captures label agreement among immediate neighbors, our definition of label consistency is more general. It measures label agreement among structurally related nodes across multiple levels—including multi-hop neighborhoods, clusters, and global contexts—rather than being limited to local connections. This broader perspective enables a better understanding of complex interaction patterns beyond simple homophily.

---

> > ### Comment · Reviewer_rvAf · 2025-08-05
> >
> > Thank you for answering my questions. I have no further questions and would like to retain my original score.

---

### Official Review · Reviewer_BBQq · 2025-06-29

**Clarity:** 3
**Significance:** 4
**Originality:** 3
**Rating:** 5
**Confidence:** 5

**Summary:**

The authors propose a unified framework based on hierarchical masks, demonstrating that existing Graph Transformers are implicitly designed to model hierarchical interactions, and that various interaction types can be effectively captured through the design of appropriate attention masks. Under this framework, they conduct a theoretical analysis of the node classification task and identify a core design principle for Graph Transformers. Guided by this principle, the authors develop M$^3$DPhormer, which integrates theoretically grounded mask designs, a bi-level attention Mixture-of-Experts mechanism, and a dual attention computation scheme. Extensive experiments are conducted to validate the effectiveness of the proposed method.

**Questions:**

In the standard Transformer architecture, the FFN module is essential for enhancing representational capacity. However, the proposed method seems to omit this component. What is the motivation of this modification?

**Ethical Concerns:**

["NO or VERY MINOR ethics concerns only"]

**Final Justification:**

After reviewing the authors’ response and the discussions among other reviewers, all of my questions is resolved. I will maintain my score.

**Limitations:**

Yes

**Quality:**

4

**Strengths And Weaknesses:**

Strengths:
1. The authors propose a unified framework from the perspective of hierarchical masks, which reduces the design space to the construction of attention masks, avoiding reliance on intricate architectural designs.
2. The authors develop a novel theoretical framework to analyze the distinct roles of various types of interactions and establish a theoretically grounded design principle that can guide future research in the field. The theoretical analysis is comprehensive, and the proofs are rigorous and well-founded.
3. The proposed method M$^3$DPhormer is novel. Based on theoretical insights, the authors design three hierarchical attention masks and introduce a bi-level expert attention routing mechanism, where each MHA module with a distinct attention mask is treated as an expert. This allows the model to adaptively select the most informative interaction for each node. Additionally, a novel dual attention computation scheme is presented to improve computational efficiency.
4. Comprehensive experiments are conducted to validate the effectiveness of the proposed approach, demonstrating its advantages over existing methods.
Weaknesses:
Overall, the paper is strong. One minor drawback is the lack of discussion on graph-level tasks, which could further demonstrate the generality of the proposed framework.

---

> ### Author Rebuttal · Authors · 2025-07-30
>
> Thank you for your valuable and positive feedback on our work. I’m glad to share our observations regarding the role of the Feed-Forward Network (FFN) module in node classification tasks.
>
> **The impact of  FFN module**
>
> Our experiments reveal that, unlike in NLP and CV domains, the commonly used FFN module does not contribute positively to performance in node classification. The following table presents comparative results across seven benchmark datasets:
>
> |      | cora       | citeseer   | pubmed     | photo      | computer   | chameleon  | squirrel   |
> | ---- | ---------- | ---------- | ---------- | ---------- | ---------- | ---------- | ---------- |
> | our  | 88.48±1.94 | 77.53±1.56 | 89.96±0.49 | 95.91±0.68 | 92.09±0.46 | 47.09±4.05 | 44.34±1.94 |
> | +ffn | 86.50±1.87 | 75.53±2.19 | 89.28±0.3  | 95.37±0.54 | 91.28±0.72 | 44.22±2.81 | 41.36±1.82 |
>
> As shown, adding the FFN module often leads to a noticeable performance drop—especially on datasets such as cora, citeseer, chameleon, and squirrel. We speculate that this is due to the nature of node features in these datasets, which are generally simple and easy to learn. In such cases, capturing complex node interactions becomes more crucial, a task better handled by the Multi-Head Attention (MHA) module. While the MHA is often regarded as the core of Transformer architectures, it is worth noting that the standard FFN module typically contains twice as many parameters as the MHA (e.g., with a projection from dimension $d \rightarrow 4d \rightarrow d$). Since meaningful structural interactions are primarily modeled through attention, a natural design choice is to allocate more capacity to the MHA module instead.
>
> In addition to the observed performance improvement, reducing or simplifying the FFN module can significantly enhance computational and memory efficiency, making the model more lightweight and scalable.

---

> > ### Comment · Reviewer_BBQq · 2025-08-06
> >
> > I have carefully reviewed the authors’ rebuttal, and all of my previous questions have been clearly addressed.

---

### Official Review · Reviewer_Exu6 · 2025-07-01

**Clarity:** 3
**Significance:** 3
**Originality:** 2
**Rating:** 4
**Confidence:** 4

**Summary:**

This paper introduces M3Dphormer, a graph Transformer architecture, and a framework for modeling graph interactions at multiple levels (local, cluster, global) through hierarchical attention masks. The central premise is that various Graph Transformer (GT) designs can be interpreted as applying different attention mask patterns. The authors unify these under a single masking framework and provide a theoretical analysis suggesting that classification accuracy correlates with a node’s receptive field size and the label consistency of its neighborhood. This leads to the design principle that an effective attention mask should cover a large, label-homogeneous neighborhood. Arguing that no single mask type is optimal across all nodes and graphs, the authors propose combining hierarchical masks.

The paper then presents M3Dphormer, a Mixture-of-Experts (MoE) based graph Transformer that integrates three mask types (local, cluster, global). The model uses a bi-level gating mechanism to weight expert modules, each associated with a specific mask type. To manage computational costs on large graphs, it employs a dual-mode attention computation that switches between dense and sparse implementations based on mask sparsity. The authors report experimental results on nine benchmark datasets, comparing M3Dphormer against 15 baselines. The model is shown to achieve strong performance across these datasets.

The main contributions are: first, a unified hierarchical mask framework that categorizes existing GT architectures; second, a theoretical analysis linking receptive field size and neighborhood label consistency to classification accuracy; third, the M3Dphormer model, which features multi-level masks, a bi-level MoE routing mechanism, and a dual-mode attention for scalability; and fourth, an empirical evaluation demonstrating performance gains over baseline models.

**Questions:**

Ablation on Global Nodes: An ablation study comparing the label-informed global nodes to generic, unlabeled global tokens (as in Exphormer) would be valuable. This would quantify the benefit of injecting supervised signals into the global attention mechanism.

Runtime Performance: Including training time or throughput metrics would provide a more complete picture of the model's practical efficiency alongside pre existing memory usage analysis.

**Ethical Concerns:**

["NO or VERY MINOR ethics concerns only"]

**Limitations:**

Scope of Evaluation: The evaluation is confined to node classification. While the results are strong, this narrow focus leaves open questions about the framework's generalizability. The performance gains on standard homophilic benchmarks (e.g., Pubmed, Ogbn-Arxiv) are modest, suggesting the model's complexity may not be justified in all scenarios. It would also be informative to compare against a simpler integration of the masks, such as concatenating them and allowing a standard attention mechanism to learn the weights, to better isolate the benefit of the explicit bi-level gating.

Limited Novelty of Components: The novelty is somewhat limited, as individual components are adapted from prior work. The use of virtual nodes for clusters and global tokens has appeared in models like GraphViT, Cluster-GT, and Exphormer. The Mixture-of-Experts (MoE) gating mechanism is also an established technique in GNNs (e.g., GCN-MoE, MOWST). While the specific combination of three expert types within a bi-level gating framework for GTs appears new, the main contribution lies in the integration and empirical validation of these components rather than a fundamentally new algorithmic paradigm.


Weaknesses related to component-level novelty and a narrow task focus still remain.

**Paper Formatting Concerns:**

Not exactly focused on formatting issues while reviewing.

**Quality:**

3

**Strengths And Weaknesses:**

Strengths

Model Design (M3Dphormer): The model architecture directly addresses the issues identified in the analysis. By using separate attention "experts" for local, cluster, and global interactions, combined with a bi-level gating mechanism to weight them on a per-node basis, M3Dphormer can adaptively combine information from different scales. The hierarchical nature of the gating—first deciding between local and non-local, then between cluster and global—is a reasonable approach to prioritize local signals while incorporating broader context.

Scalability via Dual-Mode Attention: The introduction of a dual-mode attention computation is a practical component for scalability. Since graph attention masks often have irregular sparsity, standard dense attention can be inefficient. The model switches between sparse and dense attention based on a derived criterion (Proposition 4.2) to optimize computation. This results in a significant reduction in memory usage.

Empirical Performance: M3Dphormer achieves strong empirical results on a range of benchmarks, including both homophilic and heterophilic graphs. The performance improvements are most notable on heterophilic datasets, where multi-scale information is often critical. The model is benchmarked against a comprehensive set of baselines, including recent GTs and tuned GNNs, which strengthens the empirical claims. Ablation studies in Table 3 demonstrate the contribution of each component of the proposed model.

Clarity and Organization: The paper is generally well-structured and clearly written. The hierarchical mask framework is introduced with helpful examples (Figure 1) and a clear mapping to prior work (Table 4), which contextualizes the contributions.

Weaknesses

Theoretical Assumptions and Generality: The analysis is based on a simplified generative model. The class-conditional Gaussian assumption, with orthogonal mean vectors and isotropic variance for each class, is a strong simplification of real-world feature distributions. Furthermore, the model design, particularly the global mask, injects label information from the training set by creating global virtual nodes for each class. This tailors the model to semi-supervised node classification, and its applicability to other tasks (e.g., graph classification, link prediction, or settings with unseen classes) is not demonstrated.

---

> ### Author Rebuttal · Authors · 2025-07-30
>
> We sincerely thank the reviewer for the precious time spent reading through the paper and giving constructive suggestions. Below, we address the reviewer's concerns one by one, hoping that a better understanding of every point can be delivered.
>
> **Theoretical assumptions and generality**
>
> We thank the reviewer for the insightful comment regarding the assumptions in our theoretical analysis. We acknowledge that the use of class-conditional isotropic Gaussian distributions with orthogonal class means is a simplification and may not fully capture the complexity of real-world feature distributions. However, such assumptions are widely adopted in theoretical studies of machine learning and representation learning [1,2], as they enable analytically tractable results and offer conceptual clarity. Our goal is not to precisely mimic empirical data distributions, but to distill key factors that influence node classification and to provide theoretical grounding for the design of Graph Transformers.
>
> Importantly, our analysis is conducted under the proposed unified hierarchical mask framework, which offers a feasible way for analyzing multi-level interactions. We believe this framework lays the foundation for more refined theoretical studies in the future, potentially under more relaxed or realistic distributional assumptions.
>
> **Extension of our method to graph-level task**
>
> The label-related global mask in our model is derived from theoretical analysis tailored for node classification. To extend our method to other tasks, we replace this mask with a general global mask, similar to that used in Exphormer[3]. Specifically, we adapt our model for graph-level tasks by incorporating edge features and Laplacian positional encodings, along with the general global mask.
>
> We evaluate this extension on two graph classification datasets: OGBG-MOLBACE and OGBG-MOLBBBP. Due to time constraints, we compare our method with three widely recognized GNN baselines—GCN, GAT, and GINE—known for their strong performance on graph-level benchmarks [4]. The results are presented below:
>
> |       | ogbg-bace(ROC-AUC) | ogbg-bbbp(ROC-AUC) |
> | ----- | ------------------ | ------------------ |
> | our   | 0.80432±0.010395   | 0.68232±0.006317   |
> | gcn*  | 0.75680±0.016756   | 0.65146±0.011839   |
> | gat*  | 0.78149±0.01900    | 0.65175±0.012214   |
> | gine* | 0.74799±0.010136   | 0.65095±0.011434   |
>
> These results demonstrate that our method generalizes well to graph-level tasks and outperforms strong baselines, highlighting its flexibility and effectiveness beyond node classification.
>
> **More ablation studies**
>
> We thank the reviewer for the constructive suggestions. To further validate our design choices, we conducted two additional ablation studies: (1) replacing label-informed global nodes with generic, unlabeled ones, and (2) simplifying the hierarchical mask integration by directly concatenating the three masks and applying standard attention. The results are summarized below:
>
> |                      | cora       | citeseer   | pubmed     | photo      | computer   | chameleon  | squirrel   |
> | -------------------- | ---------- | ---------- | ---------- | ---------- | ---------- | ---------- | ---------- |
> | our                  | 88.48±1.94 | 77.53±1.56 | 89.96±0.49 | 95.91±0.68 | 92.09±0.46 | 47.09±4.05 | 44.34±1.94 |
> | generic global nodes | 88.24±1.70 | 75.80±1.33 | 89.67±0.30 | 95.49±0.43 | 91.76±0.47 | 43.68±4.73 | 42.26±1.91 |
> | mask_concat          | 84.05±1.71 | 75.58±1.68 | 89.22±0.36 | 94.55±0.96 | 91.35±0.77 | 44.13±2.30 | 40.18±2.05 |
>
> We observe that both variants lead to consistent performance drops, particularly on heterophilic benchmarks (e.g., chameleon and squirrel). For the global nodes, this degradation aligns with our theoretical analysis: generic global masks lack class-specific semantics, limiting their ability to guide meaningful aggregation. In contrast, label-informed global nodes can better capture class-dependent representations and facilitate more targeted global interactions (see Section 4.2).
>
> Similarly, the mask concatenation variant suffers due to semantic ambiguity—once the masks are merged, the model can no longer distinguish whether an edge reflects local, cluster-level, or global structure. As discussed in Section 3, these interaction levels play distinct roles depending on the graph topology. This highlights the importance of disentangled hierarchical modeling and our bi-level attention MoE, which explicitly separates and selectively leverages structural signals at different scales. Simpler alternatives fail to retain this structural granularity, resulting in reduced effectiveness.
>
> **Run time performance**
>
> We are pleased to provide a detailed time complexity analysis. For standard multi-head attention (MHA), the computational cost primarily arises from: (1) feature transformations of the QKV matrices ($O(3Nd^2)$), (2) computation of the attention matrix ($O(N^2d)$), and (3) multiplication between the attention matrix and the value matrix ($O(N^2d)$). Thus, the overall time complexity is $O(3Nd^2 + 2N^2d)$. For sparse MHA, the complexity includes: (1) QKV transformations ($O(3Nd^2)$), (2) sparse attention score computation ($O(2md)$, step 4 of Algorithm 2), and (3) output aggregation ($O(2md)$, step 6 of Algorithm 2), leading to a total complexity of $O(3Nd^2 + 4md)$. A direct comparison reveals that sparse MHA is more efficient than standard MHA when $\frac{m}{N^2} < \frac{1}{2}$.
>
> In our "Dual Attention Computation Scheme" (Section 4.2), we apply sparse attention in most regions, except for the attention from global nodes to the origin node, which uses a dense pattern (where $\frac{m}{N^2} = 1$; see Figure 1). This hybrid design enables our method to achieve lower overall time complexity than both standard and sparse MHA, i.e., $\min(O(3Nd^2 + 2N^2d), O(3Nd^2 + 4md))$. To empirically validate efficiency, we report training times (over 200 epochs) on seven datasets, comparing our method, its sparse/dense variants, and PolyNormer [5]:
>
> |                       | cora       | citeseer   | pubmed     | photo      | computer   | chameleon  | squirrel   |
> | --------------------- | ---------- | ---------- | ---------- | ---------- | ---------- | ---------- | ---------- |
> | base                  | 88.48±1.94 | 77.53±1.56 | 89.96±0.49 | 95.91±0.68 | 92.09±0.46 | 47.09±4.05 | 44.34±1.94 |
> | base_time_200_epoch   | 5.62s      | 4.92s      | 6.41s      | 5.34s      | 10.86s     | 5.58s      | 12.29s     |
> | dense_time_200_epoch  | 6.69s      | 7.28s      | OOM        | 7.25s      | OOM        | 6.28s      | 16.45s     |
> | sparse_time_200_epoch | 6.28s      | 5.67s      | 7.16s      | 6.01s      | 15.56s     | 7.60s      | 14.38s     |
> | polynormer            | 3.59s      | 2.76s      | 3.60s      | 6.02s      | 4.16s      | 3.88s      | 5.53s      |
>
> Our method consistently outperforms its dense and sparse MHA variants in training efficiency.  Although slightly slower than Polyformer, the use of three MHA experts leads to faster and more stable convergence, ultimately resulting in higher accuracy. Furthermore, since the model converges within 200 epochs on most datasets, the additional overhead is acceptable in practice.
>
> **Clarifying the limitations**
>
> We acknowledge that the performance gains on standard homophilic benchmarks (e.g., Pubmed, ogbn-arxiv) are relatively modest. This is consistent with our theoretical analysis: in highly homophilic graphs, local neighborhood information already provides strong predictive signals, making complex hierarchical modeling less critical. Nonetheless, our method still consistently outperforms strong baselines across all nine datasets and exhibits stable training behavior (see Figure 5 and Appendix H.3). More importantly, our framework is designed to address the challenges of heterophilic or structurally diverse graphs, where multi-level dependencies are essential. On such datasets (e.g., Chameleon, Squirrel, Minesweeper), our model shows substantial improvements, highlighting its strength in capturing complex structural signals.
>
> Regarding novelty, we emphasize that our key contribution lies not only in the specific model, but in the unified hierarchical mask framework. This framework provides a principled and general perspective to reinterpret GTs and enables theoretical analysis of node classification. It also facilitates a extensible way to incorporate hierarchical interactions via mask design, avoiding ad hoc architecture engineering. Our method is a natural instantiation of this framework, guided by both theory and empirical findings. In addition, to the best of our knowledge, this is the first work to integrate the MoE paradigm into Graph Transformers. Unlike prior approaches in NLP and GNNs that define experts as distinct parameter sets or layers, we treat each MHA module—coupled with a specific hierarchical mask—as a relational expert. This introduces a new form of structure-aware specialization, representing a conceptual shift in how MoE mechanisms can be utilized in graph representation learning.
>
> [1] Shen, Ruoqi, et al. "Data augmentation as feature manipulation." *International conference on machine learning*. PMLR, 2022.
>
> [2] Li, Yibo, et al. "Graph fairness learning under distribution shifts." *Proceedings of the ACM Web Conference 2024*. 2024.
>
> [3] Shirzad, Hamed, et al. "Exphormer: Sparse transformers for graphs." *International Conference on Machine Learning*. PMLR, 2023.
>
> [4] Luo, Yuankai, et al. "Can Classic GNNs Be Strong Baselines for Graph-level Tasks? Simple Architectures Meet Excellence." *arXiv preprint arXiv:2502.09263* (2025).
>
> [5] Deng, Chenhui, et al. "Polynormer: Polynomial-expressive graph transformer in linear time." *arXiv preprint arXiv:2403.01232* (2024).

---

> > ### Comment · Reviewer_Exu6 · 2025-08-08
> >
> > I thank the authors for answering my questions. I have more questions to ask. Would like to keep my scores.

---

### Official Review · Reviewer_zo8h · 2025-07-02

**Clarity:** 2
**Significance:** 2
**Originality:** 2
**Rating:** 4
**Confidence:** 3

**Summary:**

This paper proposes a unified framework that reconceptualizes various Graph Transformer architectures as the design of hierarchical attention masks, leading to a key design principle: an effective mask must balance a large receptive field with high label consistency. Guided by this theory, the authors introduce M³I, a novel Mixture-of-Experts-based model that adaptively combines local, cluster, and global interaction masks to achieve state-of-the-art results on multiple node classification benchmarks.

**Questions:**

See Weaknesses. If the more convincing experimental results can be provided, I might consider increasing the score.

**Ethical Concerns:**

["NO or VERY MINOR ethics concerns only"]

**Final Justification:**

The author addressed most concerns

**Quality:**

3

**Strengths And Weaknesses:**

Strengths:
1. This paper is well-organized.
2. There is comprehensive theoretical analysis.
3. The performance is great.

Weaknesses:
1. The proposed method does not truly achieve state-of-the-art performance. The results reported in papers such as https://arxiv.org/pdf/2410.02296 and https://arxiv.org/pdf/2410.13798 are competitive. More baselines should be included for comparison.
2. The experimental evaluation is insufficient. Additional experiments, such as analyses of time and space complexity, should be conducted.
3. The paper claims that the code is publicly available, but the code for the proposed method has not been released.
4 .The authors only evaluate performance on node classification tasks, which limits the persuasiveness of the proposed method.

---

> ### Author Rebuttal · Authors · 2025-07-30
>
> We are truly grateful to the reviewer for the constructive comments and valuable feedback regarding our paper. To thoroughly address all the concerns raised, we will provide detailed responses to each question sequentially.
>
> **Compare with recent works**
>
> We sincerely thank the reviewer for highlighting these recent works. We have carefully reviewed both papers and acknowledge that they are high-quality contributions to the field.
>
> The first paper[1] proposes a novel approach that integrates large language models (LLMs) with graph neural networks (GNNs) to achieve impressive node classification performance. While the results are indeed strong, we believe this method is not a suitable baseline for our comparison. The approach leverages pre-trained LLMs, which inherently encode extensive world knowledge, and utilizes textual attributes as input features. These text-based features are significantly more informative than the original node features (which are typically simple bag-of-words or structural encodings) used in our setting. Therefore, the problem formulation and input assumptions differ substantially from ours, making a direct comparison less meaningful.
>
> The second paper [2] proposes a compelling approach that leverages Residual VQ-VAE to learn expressive token representations and employs node sampling strategies to generate transformer-compatible sequences. We have acknowledged this work in the Introduction, highlighting it as a representative method of the second line of Graph Transformer research. During the rebuttal phase, we conducted a comprehensive comparison with GQT on seven node classification benchmarks. Our model consistently outperforms GQT across all datasets, as summarized below:
>
> |      | cora       | citeseer   | pubmed     | photo      | computer   | chameleon  | squirrel   |
> | ---- | ---------- | ---------- | ---------- | ---------- | ---------- | ---------- | ---------- |
> | our  | 88.48±1.94 | 77.53±1.56 | 89.96±0.49 | 95.91±0.68 | 92.09±0.46 | 47.09±4.05 | 44.34±1.94 |
> | GQT  | 86.68±1.96 | 75.63±1.87 | 86.51±0.18 | 94.39±0.68 | 90.12±0.06 | 41.26±3.25 | 36.37±2.43 |
>
> **Note:** Although the authors have released their code on GitHub, we found the implementation to be incomplete and containing several bugs. We made our best effort to complete the missing parts and fix the bugs. However, their paper does not provide detailed hyperparameter settings per dataset. Given the time constraints, we conducted a reasonable grid search to tune hyperparameters, but were unable to fully reproduce the performance reported in their paper. We plan to contact the authors for further clarification and will update our comparison accordingly in a future version of the manuscript.
>
> **Time and space complexity**
>
> Thanks for your valuable suggestions. We have conducted comprehensive space complexity analysis for standard MHA and sparse MHA in Appendix B. The conclusion is that the total space complexity of standard MHA is $O(2hN^2)$, where $h$ is the number of heads and $N$ is the number of nodes, and the overall space complexity of sparse MHA is $O(6mhd_h)$, where $m$ is the number of non-zero elements in the mask matrix and $d_h$ is the hidden size of each head. According to our proposed "Dual Attention Computation Scheme" and Proposition 4.2, the space complexity of our method is the lower bound of standard MHA and sparse MHA, i.e., $\min(O(2hN^2), O(6mhd_h))$. We have plotted the memory usage of our method and its variants in Figure 3 and Appendix H. The numerical comparison is as follows:
>
> |        | cora    | citeseer | pubmed  | computer | photo   | squirrel | chameleon | minesweeper | arxiv   |
> | ------ | ------- | -------- | ------- | -------- | ------- | -------- | --------- | ----------- | ------- |
> | Dual   | 516 MB  | 552 MB   | 1298 MB | 1986 MB  | 794 MB  | 556 MB   | 1760 MB   | 1700 MB     | 18.9 GB |
> | Sparse | 610 MB  | 648 MB   | 1580 MB | 2364 MB  | 874 MB  | 1980 MB  | 590 MB    | 1900 MB     | OOM     |
> | Dense  | 2108 MB | 3944 MB  | OOM     | OOM      | 5062 MB | 5476 MB  | 618 MB    | OOM         | OOM     |
>
> We find that our "Dual Attention Computation Scheme" is more memory-efficient than both the Sparse MHA variant and the Dense MHA variant, demonstrating the effectiveness of our design.
>
> Now, we analyze the time complexity. For the standard MHA, the time complexity mainly comes from the feature transformations of the QKV matrices ($O(3Nd^2)$), the computation of the attention matrix ($O(N^2d)$), and the matrix multiplication between the attention matrix and the value matrix ($O(N^2d)$). Therefore, the total time complexity of standard MHA is $O(3Nd^2 + 2N^2d)$. For sparse MHA, the time complexity includes the feature transformations ($O(3Nd^2)$), the sparse attention score computation ($O(2md)$, step 4 of Algorithm 2), and the output computation ($O(2md)$, step 6 of Algorithm 2), resulting in a total of $O(3Nd^2 + 4md)$. A straightforward calculation shows that sparse MHA is more efficient than standard MHA when $\frac{m}{N^2} < \frac{1}{2}$.
>
> According to our "Dual Attention Computation Scheme" and Proposition 4.2, we apply sparse attention computation in most regions, except for attention from global nodes to the origin node, which involves a dense attention region (where $\frac{m}{N^2} = 1$, as shown in Figure 1). This hybrid design allows our method to achieve lower overall time complexity than both standard and sparse MHA, i.e., $\min(O(3Nd^2 + 2N^2d), O(3Nd^2 + 4md))$.
>
> We compare the training time (200 epochs) of our method with its variants and PolyNormer[3]. The results are as follows:
>
> |                       | cora       | citeseer   | pubmed     | photo      | computer   | chameleon  | squirrel   |
> | --------------------- | ---------- | ---------- | ---------- | ---------- | ---------- | ---------- | ---------- |
> | base                  | 88.48±1.94 | 77.53±1.56 | 89.96±0.49 | 95.91±0.68 | 92.09±0.46 | 47.09±4.05 | 44.34±1.94 |
> | base_time_200_epoch   | 5.62s      | 4.92s      | 6.41s      | 5.34s      | 10.86s     | 5.58s      | 12.29s     |
> | dense_time_200_epoch  | 6.69s      | 7.28s      | OOM        | 7.25s      | OOM        | 6.28s      | 16.45s     |
> | sparse_time_200_epoch | 6.28s      | 5.67s      | 7.16s      | 6.01s      | 15.56s     | 7.60s      | 14.38s     |
> | polynormer            | 3.59s      | 2.76s      | 3.60s      | 6.02s      | 4.16s      | 3.88s      | 5.53s      |
>
> We observe that our "Dual Attention Computation Scheme" achieves better training speed than both its sparse and dense MHA variants. Although it is slightly slower than PolyNormer, the extra computation stems from the use of three MHA experts, which are essential for achieving superior accuracy. Moreover, 200 epochs are sufficient for convergence on most datasets, making the time overhead acceptable.
>
> **Code**
>
> We have included an anonymous ZIP archive of our code in the Supplementary Material. The results can be reproduced by running the provided runs.sh script. Upon acceptance of the paper, we will make the complete codebase publicly available on GitHub.
>
> **Performance on graph-level tasks**
>
> We extend our method to support graph-level tasks by incorporating edge features and Laplacian positional encodings. To evaluate the effectiveness of this extension, we conduct experiments on two graph classification datasets: OGBG-MOLBACE and OGBG-MOLBBBP. Due to time constraints, we compare our method against three widely-recognized GNN baselines—GCN, GAT, and GINE—which have been shown to be strong baselines on graph-level benchmarks [4]. The results are summarized in the following table:
>
> |       | ogbg-bace(ROC-AUC) | ogbg-bbbp(ROC-AUC) |
> | ----- | ------------------ | ------------------ |
> | our   | 0.80432±0.010395   | 0.68232±0.006317   |
> | gcn*  | 0.75680±0.016756   | 0.65146±0.011839   |
> | gat*  | 0.78149±0.01900    | 0.65175±0.012214   |
> | gine* | 0.74799±0.010136   | 0.65095±0.011434   |
>
> As shown in the table, our extended model outperforms the baselines on both datasets, demonstrating its strong potential for graph-level tasks.
>
> [1] Xu, Zhe, et al. "How to make LLMs strong node classifiers?." *arXiv preprint arXiv:2410.02296* (2024).
>
> [2] Wang, Limei, et al. "Learning graph quantized tokenizers." *arXiv preprint arXiv:2410.13798* (2024).
>
> [3] Deng, Chenhui, et al. "Polynormer: Polynomial-expressive graph transformer in linear time." *arXiv preprint arXiv:2403.01232* (2024).
>
> [4] Luo, Yuankai, et al. "Can Classic GNNs Be Strong Baselines for Graph-level Tasks? Simple Architectures Meet Excellence." *arXiv preprint arXiv:2502.09263* (2025).

---

> > ### Comment · Reviewer_zo8h · 2025-08-04
> >
> > Thanks for your clarification. I will increase my score to 4

---

### Comment · Area_Chair_qitj · 2025-08-08

Dear Reviewers,

Thank you for your reviews. The authors have submitted their rebuttal, please take a moment to indicate whether your concerns have been adequately addressed, and update your assessment if needed.

Best,

AC

---

### Note · Authors · 2025-08-12

**Dear AC & Reviewers**

Thank you for the positive assessment and valuable feedback. We are encouraged by the recognition of our main contributions, including the novelty, strong theoretical foundations, sound techniques, and strong empirical performance.

Several common concerns were raised by the reviewers, which we have addressed as follows:

1. Generalization to graph-level tasks — In response to the comments from Reviewers zo8h and Exu6, we extended our method to support graph-level tasks by incorporating edge features and Laplacian positional encodings. The superior performance of this extended version on graph classification demonstrates the flexibility and effectiveness of our approach beyond node classification.
2. Time complexity and runtime efficiency — Also in response to zo8h and Exu6, we provided detailed theoretical analysis and experimental evidence showing that our dual-attention computation scheme is more efficient than the standard MHA module.

In addition to addressing these shared concerns, we incorporated other constructive suggestions, including: (i) comparing with more recent baselines (suggested by zo8h), (ii) conducting more extensive ablation studies to assess the contribution of individual components—such as label-informed global masks, bi-level attention MoE, and the removal of the FFN module (suggested by Exu6 and BBQq), and (iii) exploring parameter sharing among the three experts to improve efficiency (suggested by rvAf).

Following our rebuttal, reviewer zo8h, who initially gave the only negative score, has confirmed their intention to raise the score to 4. Other reviewers with positive initial scores indicated that their concerns were resolved, and no new issues were raised during the discussion stage.

We hope this information will assist the AC in making the final decision.

Sincerely,
Authors

---

### Decision · Program_Chairs · 2025-09-17

**Decision:**

Accept (poster)

**Comment:**

This paper proposes a unified hierarchical mask framework for Graph Transformers and introduces a Mixture-of-Experts-based model that adaptively integrates local, cluster, and global attention masks for node classification. The method is theoretically well-motivated, clearly written, and demonstrates strong empirical performance across multiple benchmark datasets, particularly on heterophilic graphs. Most reviewers acknowledged the contributions in terms of methodological novelty, theoretical analysis, and adaptive multi-level fusion, as well as the practical scalability achieved via dual-mode attention computation. At the same time, several reviewers suggested improvements regarding generalizability and reproducibility, including comparisons with more recent baselines, evaluation beyond node classification, analysis of computational cost, and release of code. In response to these comments, the authors provided clarifications on the model design, detailed the cluster-level mask construction and gating mechanism, and highlighted plans for code release and extended experiments. Considering its theoretical contribution, strong empirical results, and responsive discussion of reviewer concerns, I recommend accepting this paper, with potential for broader application in future work.